# 5-Azacytidine Downregulates the Proliferation and Migration of Hepatocellular Carcinoma Cells In Vitro and In Vivo by Targeting miR-139-5p/ROCK2 Pathway

**DOI:** 10.3390/cancers14071630

**Published:** 2022-03-23

**Authors:** Federica Tonon, Maja Cemazar, Urska Kamensek, Cristina Zennaro, Gabriele Pozzato, Sergio Caserta, Flora Ascione, Mario Grassi, Stefano Guido, Cinzia Ferrari, Laura Cansolino, Francesco Trotta, Biljana Grcar Kuzmanov, Giancarlo Forte, Fabiana Martino, Francesca Perrone, Riccardo Bomben, Valter Gattei, Nicola Elvassore, Erminio Murano, Nhung Hai Truong, Michael Olson, Rossella Farra, Gabriele Grassi, Barbara Dapas

**Affiliations:** 1Department of Life Sciences, Cattinara University Hospital, Trieste University, Strada di Fiume 447, I-34149 Trieste, Italy; ftonon@units.it (F.T.); fp385@cam.ac.uk (F.P.); bdapas@units.it (B.D.); 2Department of Experimental Oncology, Institute of Oncology Ljubljana, Zaloska 2, SI-1000 Ljubljana, Slovenia; mcemazar@onko-i.si (M.C.); UKamensek@onko-i.si (U.K.); BKuzmanov@onko-i.si (B.G.K.); 3Faculty of Health Sciences, University of Primorska, Polje 42, SI-6310 Izola, Slovenia; 4Department of Medical, Surgical and Health Sciences, University of Trieste, Cattinara Hospital, Strada di Fiume 447, I-34149 Trieste, Italy; cristina.zennaro@asugi.sanita.fvg.it (C.Z.); gabriele.pozzato@asugi.sanita.fvg.it (G.P.); 5Department of Chemical, Materials and Industrial Production Engineering, University of Naples “Federico II”, Piazzale V. Tecchio 80, I-80125 Naples, Italy; sergio.caserta@unina.it (S.C.); floriana.ascione@gmail.com (F.A.); stefano.guido@unina.it (S.G.); 6CEINGE Advanced Biotechnologies, via Gaetano Salvatore, 486, I-80145 Napoli, Italy; 7Department of Engineering and Architecture, University of Trieste, Via Valerio 6/A, I-34127 Trieste, Italy; MARIO.GRASSI@dia.units.it; 8Department of Clinic-Surgical Sciences, Laboratory of Experimental Surgery and Animal Facility, University of Pavia, Via Ferrata 9, I-27100 Pavia, Italy; cinzia.ferrari@unipv.it (C.F.); lauracansolino@libero.it (L.C.); 9Department of General Surgery, Maggiore Hospital, Largo Donatori del Sangue 1, I-26900 Lodi, Italy; ceccotrotta@libero.it; 10International Clinical Research Center (ICRC) of St Anne’s University Hospital, CZ-65691 Brno, Czech Republic; giancarlo.forte@fnusa.cz (G.F.); fabianamartino@outlook.it (F.M.); 11Department of Paediatrics, University of Cambridge, Addenbrooke’s Hospital, Hills Road, Cambridge CB2 0QQ, UK; 12Clinical and Experimental Onco-Haematology Unit, Centro di Riferimento Oncologico, Istituto di Ricovero a Cura a Carattere Scientifico IRCCS, 33081 Aviano, Italy; rbomben@cro.it (R.B.); vgattei@cro.it (V.G.); 13Industrial Engineering Department, University of Padova, Via Francesco Marzolo, 9, I-35131 Padova, Italy; nicola.elvassore@unipd.it; 14Nealys SRL, Via Flavia 23/1, I-34148 Trieste, Italy; murano@nealys.com; 15Stem Cell Research and Application Laboratory, VNUHCM, University of Science, Ho Chi Minh City 72711, Vietnam; thnhung@hcmus.edu.vn; 16Department of Chemistry and Biology, X University, MaRS Discovery District, West Tower 661 University Avenue, Toronto, ON M5G 1M1, Canada; michael.olson@ryerson.ca

**Keywords:** hepatocellular carcinoma, 5-azacytidine, cell cycle, migration, miR-139-5p, ROCK2, E2F1, MMP-2

## Abstract

**Simple Summary:**

For hepatocellular carcinoma (HCC), the second most common cause of cancer-related death, effective therapeutic approaches are lacking. As aberrant gene methylation is a major contributor to the development of HCC, demethylating drugs such as 5-azacytidine (5-Aza) have been proposed. However, despite the potential efficacy of 5-Aza in HCC, most of its mechanisms of action are still unknown. Here, we investigate the phenotypic/molecular effects of 5-Aza with a focus on miR-139-5p. Using multiple in vitro and in vivo models of HCC, we show for the first time that 5-Aza can impair HCC development via upregulation of miR-139-5p, which in turn downregulates the ROCK2/cyclin D1/E2F1/cyclin B1 pro-proliferative pathway and the ROCK2/MMP-2 pro-migratory pathway. These observations elucidate the mechanisms of action of 5-Aza in HCC, strengthen its therapeutic potential, and provide novel information about the crosstalk among ROCK2/cyclin D1/E2F1/cyclin B1/MMP-2 in HCC.

**Abstract:**

Background: For hepatocellular carcinoma (HCC), effective therapeutic approaches are lacking. As aberrant gene methylation is a major contributor to HCC development, demethylating drugs such as 5-azacytidine (5-Aza) have been proposed. As most 5-Aza mechanisms of action are unknown, we investigated its phenotypic/molecular effects. Methods: 5-Aza effects were examined in the human HCC cell lines JHH-6/HuH-7 and in the rat cell-line N1-S1. We also employed a xenograft mouse model (HuH-7), a zebrafish model (JHH-6), and an orthotopic syngeneic rat model (N1-S1) of HCC. Results: 5-Aza downregulated cell viability/growth/migration/adhesion by upregulating miR-139-5p, which in turn downregulated ROCK2/cyclin D1/E2F1 and increased p27^kip1^, resulting in G1/G0 cell accumulation. Moreover, a decrease in cyclin B1 and an increase in p27^kip1^ led to G2/M accumulation. Finally, we observed a decrease in MMP-2 levels, a stimulator of HCC cell migration. Aza effects were confirmed in the mouse model; in the zebrafish model, we also demonstrated the downregulation of tumor neo-angiogenesis, and in the orthotopic rat model, we observed impaired N1-S1 grafting in a healthy liver. Conclusion: We demonstrate for the first time that 5-Aza can impair HCC development via upregulation of miR-139-5p, which in turn impairs the ROCK2/cyclin D1/E2F1/cyclin B1 pro-proliferative pathway and the ROCK2/MMP-2 pro-migratory pathway. Thus, we provide novel information about 5-Aza mechanisms of action and deepen the knowledge about the crosstalk among ROCK2/cyclin D1/E2F1/cyclin B1/p27^kip1^/MMP-2 in HCC.

## 1. Introduction

Hepatocellular carcinoma (HCC), the main form of primary liver cancer, represents the second most common cause of cancer death and the sixth most commonly diagnosed cancer worldwide [1]. Despite the development of several anti-HCC drugs, such as the multi-tyrosine kinase inhibitor sorafenib and the more recent drugs lenvatinib, regorafenib, cabozantinib, and ramucirumab, the overall survival is unacceptably short [2]. Thus, novel therapeutic options are urgently needed.

The epigenetic machinery regulates gene expression in several ways, including the control of covalent DNA modifications such as DNA methylation. A class of enzymes called DNA methyltransferases (DNMTs) [3] covalently bind a methyl group to cytosine residues within 5′-CpG-3′ palindromes [4], which are clustered in the promoters of many human genes. After methylation, proteins containing a methylcytosine binding domain [5] are recruited to the methylation sites. This, in turn, represses gene transcription by blocking access of transcription factors to the gene promoter.

Epigenetic alterations are increasingly recognized as relevant elements contributing to carcinogenesis in different types of tumors [6,7,8,9], including HCC [10]. Notably, alterations in DNA methylation have been observed to increase from cirrhosis to neoplastic lesions [11,12], leading to the downregulation of a number of different genes, including tumor suppressor genes. Aberrant methylation not only affects gene expression in HCC but can also affect the expression of microRNAs (miRs). These are double-stranded non-coding RNAs about 22 nucleotides in length that can bind to the 3′-untranslated region of the corresponding target mRNAs, leading to inhibition of protein translation and often initiating mRNA degradation [13]. A relevant miR in HCC is miR-139-5p, as its expression is significantly reduced in HCC compared to non-tumor liver tissue [14]. Moreover, miR-139-5p levels are associated with the grade/pathological stage of HCC, and its reduced expression leads to poor clinical outcome [15]. Downregulation of miR-139-5p is not only relevant in HCC, as it occurs in different tumors, such as bladder [6], colon [8], breast, lung, ovarian [9], and hematologic [7] cancers.

Among the various recognized targets of miR-139-5p is Rho-associated coiled-coil kinase-2 (ROCK2) [16]. The ROCK family of proteins [17,18], which includes ROCK1 and ROCK2, phosphorylates a variety of downstream target proteins that modify the ultrastructural assemblies of filamentous actin that are important for regulating cell contractility, motility, and morphology. Despite their similarity (64% homology in their amino acid sequence [18]), ROCK1 and ROCK2 do not have completely overlapping targets and functions [18]. Importantly, overexpression of ROCK2 correlates with poor prognosis in HCC [19], and its silencing significantly reduces HCC migration and invasion in vivo and in vitro [20,21]. 

ROCK2 is an important promoter of cell proliferation and migration. In particular, it increases the levels of cyclin D1 [22] and matrix metalloproteinase 2 (MMP-2) [19] while reducing the levels of p27^Kip1^ [22]. In the early G1 phase, cyclin D1, together with cyclin-dependent kinase 4 (CDK4), phosphorylates the retinoblastoma protein (pRB) along with cyclin E/CDK2, abolishing the repressive effect of pRB on the transcription factor E2F1 (E2 promoter binding factor) [23]. Cyclin D1 is overexpressed in HCC, and its level correlates with advanced tumor stage and progression [24,25]. E2F1 is a transcription factor that plays a role in promoting cell proliferation [26,27] and many other cellular processes, such as cell migration and differentiation. E2F1 is overexpressed in HCC [28] and is involved in the development of HCC [29]. p27^Kip1^ is a cyclin-dependent kinase inhibitor that downregulates the cell cycle by different mechanisms, including the impairment of the cyclin-dependent kinase 1 (CDK1)/cyclin B1 complex so that it can no longer promote the G2/M transition [30]. Finally, MMP-2 belongs to a family of zinc-containing proteolytic enzymes that can digest collagen IV present in the extracellular matrix, thus promoting cell migration [31]. Of note, the level of ROCK2 in HCC positively correlates with that of MMP-2 [19].

Demethylating drugs have been developed to counteract the abnormal methylation of gene promoters observed in HCC [10]. A first-generation demethylating agent is 5-azacytidine (5-Aza), an FDA-approved drug for the treatment of myelodysplastic syndromes [32]. Once incorporated into DNA, 5-Aza irreversibly binds to DNMT1 and promotes its degradation, resulting in DNA demethylation. 5-Aza can also be incorporated into RNA, which also alters gene expression [33]. In response to 5-Aza, gene expression is reactivated, and HCC cells partially recover their hepatocellular differentiation and also become more sensitive to sorafenib [34]. 

Despite the documented potential efficacy of 5-Aza in HCC, most of its mechanisms of action are still unknown. In this work, we investigate the phenotypic and molecular effects of 5-Aza with a focus on signaling pathways controlled by miR-139-5p in different in vitro and in vivo models of HCC. The focus on miR-139-5p was based on its known involvement in HCC and on the fact that 5-Aza has been documented to be able to reactivate miR-139-5p expression in different types of tumors, such as bladder [6], colon [8], breast, lung, ovarian [9], and hematologic [7] cancers. 

Our data indicate that 5-Aza can impair HCC development via upregulation of miR-139-5p, which in turn impairs the ROCK2/cyclin D1/E2F1/cyclin B1 pro-proliferative pathway and the ROCK2/MMP-2 pro-migratory pathway. Thus, we provide novel information about 5-Aza mechanisms of action and deepen the knowledge about the crosstalk among ROCK2/cyclin D1/E2F1/cyclin B1/p27^Kip1^/MMP-2 in HCC.

## 2. Materials and Methods

### 2.1. Cell Cultures

Two human HCC cell lines, HuH-7 [35] and JHH-6 [36,37], known to have intermediate and low hepatic differentiation levels, respectively, were used. As controls for non-tumorigenic liver cells, we used immortalized human hepatocytes (IHH) and hepatocyte-like cells derived from human embryonic stem cells (HLC, National Stem Cell Bank, Madison, WI, USA). JHH-6 cells were cultured in William’s medium (Sigma-Aldrich, St. Louis, MO, USA), and HuH-7 cells were grown in DMEM high-glucose medium (Euroclone, Milan, Italy). IHH cells were cultured in Ham’s Nutrient Mixture-F12 (Euroclone, Milan, Italy) containing 1μM dexamethasone (Sigma-Aldrich, St. Louis, MO, USA) and 10^−6^ μM recombinant human insulin (Humulin, Lilly, Indianapolis, IN, USA). The rat HCC cell line N1-S1 [38] was cultured in DMEM medium (Euroclone, Milan, Italy). All media contained 10% heat-inactivated fetal bovine serum (FBS) (Euroclone, Milan, Italy; Thermo Fisher Scientific, Waltham, MA, USA), 100 U/mL penicillin, 100μg/mL streptomycin, and 2 mM L-glutamine (Euroclone, Milan, Italy). All cell lines were grown at 37 °C in a humidified atmosphere at 5% CO_2_. HLC were differentiated (from human embryonic stem cells) and grown as previously reported [39]. 

### 2.2. 5-Aza Treatment

5-Azacytidine (4-amin-1-β-D-ribofuranosyl-1,3,5-trizan-2(1H)-one; 5-Aza) was provided by the pharmaceutical company Celgene as a drug powder, from which we prepared a stock solution in ultrapure sterile water. Twenty-four hours before treatment, IHH, HLC, JHH-6, HuH-7, and N1-S1 were seeded at a specific density, depending on the type of assay performed. The day after seeding, 5-Aza was administered every 24 h for three consecutive days (protocol 1, Appendix A) to mimic the repeated cycles of administration used in the clinic for hematological patients [40]. HLC, JHH-6, and IHH cells were treated with 0.8 μM or 6 μM 5-Aza. The concentration of 6 μM 5-Aza was chosen to remain within the dose range of previous publications [34,41], and based on our preliminary testing, 0.8 μM 5-Aza was chosen to mimic the average plasma concentration found in patients with myelodysplastic syndrome treated with 5-Aza [40]. HuH-7 and N1-S1 cells were also treated with a higher dose of 5-Aza (30 μM) due to their resistance to 5-Aza.

### 2.3. Cell Number, Viability, Cell Cycle, and Necrosis 

#### 2.3.1. 5-Aza Treatment 

The number of viable cells was determined using a standard trypan blue exclusion assay, followed by visualization using the Thoma counting chamber (Exacta-Optech) under a microscope (Nikon Eclipse TS100). Cell viability was determined using the MTT assay as previously described [42,43,44] by seeding 2.5 × 10^3^ HuH-7, JHH-6, and IHH and 10^3^ N1-S1 cells in 96-well microplates. Cell cycle phase assessment was performed as previously described [42,43,44]. Cell necrosis was assessed using the LDH assay kit according to the manufacturer’s instructions (BioVision Prod., Mountain View, CA, USA) [42,43,44]; protocol 1 (Appendix A) was followed for all of these assays.

#### 2.3.2. miR-139-5p Mimic/Antagomir Treatment

A miR-139-5p mimic/antagomir (Ambion from Life Technologies, Carlsbad, CA, USA) was transfected with Lipofectamine 2000 (Invitrogen, Waltham, MA, USA) into a total number of 3.6 × 10^4^ HCC cells (seeded in a 6-well plate), which were then used for cell counting, cell cycle analysis, and protein and mRNA quantification; for MTT, cells were seeded in a 96-well plate containing 4 × 10^3^ cells, as previously reported [45]. In all cases, a final miR concentration of 100 nM was used (protocol 3, Appendix A). Cell number, viability, and cell cycle were evaluated as in Section 2.3.1. When the antagomir was tested in combination with 5-Aza treatment, cells were treated with 5-Aza 6 h after completion of transfection, 5-Aza administration was repeated at 24 and 48 h, and then cells were analyzed 24 h after the last 5-Aza treatment (protocol 5, Appendix A). 

#### 2.3.3. ROCK2/E2F1 Overexpression, siROCK2, and ROCK2 Inhibitor

JHH-6/HuH-7 cells were seeded in six-well plates at a density of 1.5 × 10^5^ cells/well and 2.5 × 10^5^ cells/well, respectively. The next day, 2 μg of a plasmid expressing ROCK2 [46], a plasmid expressing E2F1 [45], or a control plasmid expressing enhanced green fluorescent protein (EGFP [45]) was transfected with Lipofectamine 2000 (Invitrogen) as previously reported [47]. One and two days later, 5-Aza was administered at a concentration of 0.8 μM or 6 μM in JHH-6 and HuH-7, respectively. Cells were collected and counted three days after transfection (protocol 6, Appendix A). When the effects of ROCK2 overexpression on E2F1 mRNA levels were tested, E2F1 mRNA levels were evaluated two days after the transfection of the ROCK2 plasmid (protocol 6 modified).

Next, siROCK2 (siGENOME SMARTpool, purchased from Dharmacon—GE Healthcare, Chicago, IL, USA) was administered at the optimized concentration of 110 nM to 3.6 × 10^4^ HCC cells seeded in a six-well plate according to protocol 3 (Appendix A), as was performed for miR-139-5p mimic/antagomir (Section 2.3.2).

For the assay with the ROCK inhibitor, JHH-6 (10^3^ cells), HuH-7 (2.5 × 10^3^ cells), and N1-S1 (2.5 × 10^3^ cells) were seeded in 96-well plates in their specific media. One day later, cells were treated with 50μM ROCK inhibitor (Y-27632; Selleckchem, Houston, TX, USA) for 4 h. Cell viability (MTT) was determined 4 h later (protocol 7, Appendix A) as described in Section 2.3.1. 

### 2.4. Cell Migration Assays

#### 2.4.1. 5-Aza Treatment 

For the scratch assay, 10 × 10^4^ JHH-6 or 20 × 10^4^ HuH-7 cells were seeded in complete medium in 3.5 cm diameter plates. Twenty-four hours after seeding (protocol 2, Figue S1), cells were treated with 5-Aza every 24 h for three consecutive days. Simultaneously with the last drug treatment, a scratch was made, and the cells were covered with their growth medium supplemented with 1% FBS and 1.25 g/mL mitomycin C (Sigma-Aldrich, St. Louis, MO, USA) to reduce proliferation stimuli. Images were taken with a microscope (Leica DM IRB, Leica, Wetzlar, Germany) immediately after scratching and 24 h afterwards. 

#### 2.4.2. miR-139-5p Mimic/Antagomir Treatment

JHH-6 cells were seeded in complete medium as described above, except that the optimal cell density was 6 × 10^4^ wells. Twenty-four hours after seeding, miR-139-5p/antagomir (100 nM) was transfected with Lipofectamine 2000. Scratching was performed the next day, and cells were grown in medium containing 1% FBS to minimize cell proliferation. Images were taken with a microscope (Leica DM IRB) immediately after the scratch was made and after 24 h (protocol 4, Appendix A). When the antagomir was tested in combination with 5-Aza, cells were treated with 5-Aza 6 h after the end of transfection and 24 and 48 h later when scratching was performed; 24 h after scratching, cells were analyzed microscopically (protocol 5, Appendix A). 

#### 2.4.3. ROCK2 Inhibitor 

HCC cells were seeded in complete medium as described in Section 2.4.1. Twenty-four hours after seeding, scratching was performed; simultaneously, cells were treated with 50μM ROCK2 inhibitor (Y-27632; Selleckchem, Houston, TX, USA) for 4 h. Thereafter, the medium containing the inhibitor was replaced with fresh medium supplemented with 1% FBS to minimize cell proliferation (protocol 7, Appendix A). Images were acquired 8 h after ROCK2 inhibitor administration.

#### 2.4.4. Bortezomib Alone or in Combination with 5-Aza

HCC cells were seeded in complete medium as in Section 2.4.1. One day after seeding, cells were treated with 10nM bortezomib alone or in combination with 0.8μM/6 μM/30 5-Aza. The 5-Aza treatment was repeated on days 2 and 3 when scratching was performed; cells were grown in medium containing 1% FBS to minimize cell proliferation (protocol 8, Appendix A). Images were taken immediately after scratching and on days 4 and 5.

#### 2.4.5. MMP-2 Inhibitor 

HCC cells were seeded in complete medium as described in Section 2.4.1. On the 2nd and 3rd days after seeding, cells were treated with 0.1μM MMP-2 inhibitor (SB-3CT; Selleckchem, Houston, TX, USA). Simultaneously with the second SB-3CT treatment, scratching was performed, and cells were grown in medium enriched with only 1% FBS to minimize cell proliferation (protocol 9, Appendix A). Images were acquired as in Section 2.4.4. 

#### 2.4.6. FATIMA and Adhesion Assays

Fluorescence-assisted transmigration, invasion, and motility assay (FATIMA) was performed as we reported previously [26]. In brief, HCC cells were treated as in protocol 1 (Appendix A), harvested, and labeled with the vital dye FAST DiI (Molecular Probes, Invitrogen). Then, 2 × 10^4^ cells were re-suspended in 150 μL of specific medium containing 0.1% BSA and added to the top of the collagen IV-coated insert, and 800μL of the specific medium containing 10% FBS and 0.1% BSA was added to the bottom of the insert. The migration of the cells was assessed hourly by spectrophotometric measurements (Infinite f200, TECAN; excitation λ = 535 nm; emission λ = 595 nm).

For the adhesion assay, JHH-6 cells were treated as described in protocol 1 (Appendix A). Then, 5 × 10^4^ cells per 100 µL of William’s medium containing 0.5% (*v*/*v*) BSA, labeled with fluorescent dye FAST DiI (1:200), were placed in a 96-well plate for 5, 10, 15, and 20 min. The wells were pre-coated with collagen I (5 µg/cm^2^). After incubation, unattached cells were removed from the wells, and the number of adherent cells was determined using Infinite f200 (TECAN, excitation λ = 535 nm; emission λ = 595 nm) based on a calibration curve consisting of an increasing number of FAST DiI-labeled cells. 

#### 2.4.7. Time-Lapse 

The kinetics of scratch closure was monitored via time-lapse imaging in JHH-6 (protocol 1, Appendix A) treated with 6 μM 5-Aza and with the addition of the proliferation inhibitor mitomycin C (1.25 μg/mL). The workstation for video microscopy [48] is based on an inverted microscope (Zeiss Axiovert 200, Carl Zeiss, Jena, Germany) housed in a homemade incubator. Using a monochromatic CCD video camera (Hamamatsu Orca AG, Hamamatsu City, Japan), multiple independent fields of view were recorded every 15 min in each cell dish with a 5X objective (EC Plan—NEOFLUAR Ph1).

Wound healing experiments were analyzed by measuring the reduction in the area of the cell-free region (A) over time, normalized with respect to baseline (A0) [48,49]. The cell-free area was measured for each time step of the time-lapse experiments using a self-developed automatic image analysis algorithm based on image variance analysis. When plotting A/A0 as a function of time, the obtained profiles showed a linear decrease after an initial time delay, all starting from A/A0 = 1, indicating that the wound healing process proceeds at a constant rate. The slope of the linear region of the A/A0 vs. t curve can be considered a measure of the wound closure rate α (h−1). 

### 2.5. Zymography Assay

After JHH-6 treatment according to protocol 1 (Appendix A), cells were collected and prepared for a standard SDS-PAGE treatment buffer under non-reducing conditions. SDS-PAGE gel contained the MMP-2 substrate, i.e., gelatin (2.8 mg/mL). After electrophoresis, SDS was removed from the zymogram by washing with the wash buffer (50 mM Tris HCl, pH 7.5, 5 mM CaCl_2_, 1 μM ZnCl, 0.02% NaN_3_, and 2.5% Triton X-100), followed by incubation in the digestion buffer (50mM Tris HCl, pH 7.5, 5 mM CaCl2, 1 μM ZnCl, and 0.02% NaN_3_) overnight at 37 °C. The zymogram was then stained with Coomassie brilliant blue, and the areas of digestion appeared as bands where the substrate was degraded by the enzyme. Since the serum contains large amounts of MMP-2, it was included as a positive control in the zymography assay. 

### 2.6. Western Blotting

Protein extraction from cell pellets, quantification, SDS-PAGE, and Western blotting were performed as previously described [44,47]. Protein extraction from animal tissue samples was performed using 20–30 mg tumor mass using RIPA buffer (supplemented with protease and phosphatase inhibitors) according to the manufacturer’s instructions (Sigma-Aldrich, St. Louis, MO, USA). The following antibodies were used: anti-E2F1 mouse monoclonal antibody (sc-251 Santa Cruz), anti-ROCK2 rabbit monoclonal antibody (ab125025 Abcam), anti-MMP-2 rabbit polyclonal antibody (sc-10736 Santa Cruz), anti-cyclin D1 mouse monoclonal antibody (GeneTex Inc., Irvine, CA, USA), anti-p27^kip1^ mouse monoclonal antibody (sc-1641 Santa Cruz), anti-cyclin B1 mouse monoclonal antibody (sc-245 Santa Cruz, Dallas, TX, USA), and anti-α-tubulin mouse monoclonal antibody (A11126, Invitrogen). On the same membrane, the loading control protein GAPDH (rabbit polyclonal antibody sc-25778 Santa Cruz) was tested. Blots were developed with the appropriate secondary horseradish peroxidase antibodies (goat anti-mouse and goat anti-rabbit antibodies, Aurogene, Rome, Italy), and chemiluminescence was enhanced with SuperSignal West Pico and West Femto substrates (Thermo Fischer Scientific, Waltham, MA, USA). Detection was performed by exposing the membranes to Kodak film (Sigma-Aldrich, St. Louis, MO, USA). Band intensities were measured using a model GS-700 Imaging Densitometer equipped with Molecular Analyst software “Quantity One” (Bio-Rad Laboratories, Hercules, CA, USA).

For the microtubule polymerization assay, HuH-7 and JHH-6 cells were cultured and treated as in protocol 1. Cells were then lysed for 10 min at room temperature with hypotonic buffer (20 mM Tris-HCl (pH 6.8), 1 mM MgCl_2_, 2 mM EGTA, and 0.5% NP40), to which phosphatase and protease inhibitors were added. The supernatant containing cytosolic tubulin was separated by centrifugation at 13,000 rpm for 10 min at room temperature. The pellet containing polymerized tubulin was collected and re-suspended in the hypotonic buffer. The lysates were loaded into 10% SDS-PAGE gels prior to Western blotting analysis. The binding conditions for the assay with anti-alpha-tubulin Ab (Invitrogen, A11126) were: dilution 1:1000 overnight at 4 °C in 5% milk and 0.1% Tween; as the secondary Ab, we used goat anti-mouse HRP, 1:12,000, 1 h at room temperature. 

### 2.7. QRT-PCR

Extraction of total RNA from cell pellets and quantification of RNA were performed as previously described [44,47]. For the zebrafish experiment, extraction of total RNA and real-time analysis were performed as described previously [50,51]. The primers (Eurofins Genomics) and real-time amplification conditions for E2F1 [43] and 28S RNA [26] have been described previously. The primers and amplification conditions for the other targets were as follows: PDE2A fw 5′ GAA AGTCCG GGA GGC TAT CAT 3′, PDE2A rev 5′ CAC TTG GGT ATC AGG AGC CA3′; ROCK2 fw 5′ GCA AAT CTG TTA ATA CTC GCC T 3′, ROCK2 rev 5′ GCT TTC CTC TGA TAT TCT GCA T 3′; albumin fw 5′ CCT GTT GCC AAA GCT CGA TG 3′, albumin rev 5′ GAAATCTCTGGCTCAGGCGA 3′. Annealing temperatures for each primer pair were 62 °C, 60 °C, and 60 °C, respectively. 

The mirVana™ miRNA Isolation Kit was used to isolate total RNA and preserve small RNAs from animal tissues and cells according to the manufacturer’s instructions (Ambion, ThermoFischer Scientific, Austin, TX, USA).

Quantification of small RNAs was performed by spectrophotometric analysis (NanoDrop ND-100; CelBio). miR-139-5p and 5s RNA reverse transcription reactions were performed using the miRCURY LNA Universal RT microRNA PCR Kit (Exiqon, Qiagen, Hilden, Germany) according to the manufacturer’s instructions. Real-time quantitative PCR conditions for miR amplification were as follows: 40 amplification cycles with pre-denaturation at 95 °C for 10 min, denaturation at 95 °C for 15 s, and annealing at 60 °C for 60 s, followed by a final dissociation phase (95 °C/60 °C/95 °C for 15 s each). The relative amounts of miR-139-5p were normalized by the 5s rRNA content. Primers for both miR and 5s were purchased from Exiqon (Qiagen, Hilden, Germany).

### 2.8. Immunofluorescence

HCC cells were seeded on slides placed in each well of 6-well plates. After specific treatment, they were fixed in 4% paraformaldehyde or in ice-cold methanol for 20 min at RT. Cells were then washed three times with 1X PBS and incubated with 0.1% Triton X- 100 at RT for 5 min to permeabilize cell membranes (only for cells fixed in PFA). Cells were then saturated in 1% BSA/1X PBS for at least 30 min and incubated with the specific primary antibody (anti-β-tubulin III antibody diluted 1:100, rabbit, Sigma-Aldrich, St. Louis, MO, USA; anti- β-actin antibody diluted 1:600, mouse, Invitrogen) overnight at 4 °C in a humid chamber. After rapid washing three times, cells were incubated with the secondary antibody (Alexa Fluor 488, 1:400, anti-rabbit, and Alexa Fluor 546, 1:400, anti-mouse, Invitrogen) for 45 min at RT in a humid chamber in the dark. Finally, cells were incubated with DAPI (1:2500 in 1X PBS; n-9564; Sigma-Aldrich, St. Louis, MO, USA) for 5 min at RT and then mounted with ProLongTM Gold Antifade reagent (Invitrogen). Images were taken using a Leica DM-2000 microscope. 

### 2.9. miR and Gene Expression

Global miR expression was assessed in untreated JHH-6 or JHH-6 treated with 6 μM 5-Aza according to protocol 1 (Appendix A). Analysis was performed by CBM (Consortium for the Center of Molecular Biomedicine, AREA Science Park, 34149, Basovizza, Trieste, Italy). Samples from two independent cell treatments were analyzed. The mirVana™ miRNA Isolation Kit (ThermoFischer Scientific) was used to isolate miRs from cells. Total RNA quality, integrity, and quantity were assessed using a Lab-on-Chip System Bioanalyzer 2100 (Applera Corporation, Norwalk, CT, USA) and a NanoDrop ND-1000 (Euroclone, Milan, Italy). Labeling was performed according to the 3DNA FlashTagTM Biotin HSR protocol. Affymetrix GeneChip miRNA arrays were used for hybridization of the labeled nucleic acid; chip readout was performed using the GeneChip Scanner 3000.

Gene expression was evaluated in samples from three independent JHH-6 cell treatments with or without 6 μM 5-Aza according to protocol 1 (Appendix A). In brief, total RNA was extracted using the RNeasy Mini Kit (Qiagen, Hilden, Germany). The quality, integrity, and quantity of total RNA were assessed spectrophotometrically as described above. Each RNA sample to be analyzed was labeled with 3-CTP cyanine using the Low Input Quick Amp Labeling Kit (Agilent Technologies). The labeled RNAs were hybridized using the Whole Human Genome Oligonucleotide Platform (4 × 44; Agilent Technologies). The hybridization signal, evaluated using multiple probes for each messenger RNA, was determined using Agilent Feature Extraction 10.7.3 software (Agilent Technologies). Bioinformatic analysis was performed using GeneSpring GX 11.0.2 software (Agilent Technologies) and the statistical t-test. The significance value (*p*) was adjusted according to the method of Benjamini and Hochberg for multiple testing, which allows multiple comparisons to be performed on independent tests. Differentially expressed genes were identified with a significance value of less than or equal to 0.05. 

### 2.10. Mathematical Modeling of 5-Aza Effects on Tumor Growth In Vivo

In order to accurately describe the effects of 5-Aza on tumor growth, a recent mathematical model aimed at describing the increase in tumor volume in vivo was used [44]. This model is based on the assumption that tumor volume, and thus the number of tumor cells (*N*), increases according to an exponential law. To account for the presence of external confounding factors (e.g., anti-tumor drugs), it is assumed that the proportion (*f*) of cells that can divide depends on time (*t*) according to the following exponential law:
(1)
$$f=\left(f_{0}{\;-\;f}_{\infty }\right)e^{{-k}_{f}t}{+f}_{\infty }$$

where *f*_0_ and *f*_∞_ are, respectively, the initial (*t* = 0) and final (*t* →∞) fractions of cells that undergo division, and *k*_f_ is a kinetic constant connected to the kinetics of decreasing *f*. Thus, *f* can be viewed as a correction factor for the classical kinetics equation describing the temporal increase in the tumor cell number *N*:
(2)
$$\frac{dN}{dt}{=k}_{D}N\left(\left(f_{0}{\;-\;f}_{\infty }\right)e^{{-k}_{f}t}{+f}_{\infty }\right)=V_{D}$$

where *k*_D_ is the kinetic constant connected to cell division speed *V_D_* (cells/time). The analytical solution of Equation (2) is:
(3)
$$\frac{N}{N_{0\;}}{=\exp(k}_{D}{((f}_{0}{\;-\;f}_{\infty })(\frac{{1-e}^{{-k}_{f}t}}{k_{f}}{)+f}_{\infty }t))$$

where *N*_0_ is the initial number of tumor cells. In data analysis, we assumed *f*_0_ = 1 (100% of the cells can undergo division) and *f*_∞_ = 0 (after a very long time, cells cannot undergo division). The combination of Equations (2) and (3) provides the expression for the velocity of tumor volume increase:
(4)
$$\frac{V_{D}}{N_{0\;}}{=V}_{D}^{+}{=k}_{D}{\;\times \;\exp(k}_{D}{((f}_{0}{-f}_{\infty })(\frac{{1-e}^{{-k}_{f}t}}{k_{f}}{)+f}_{\infty }t))\;\times \;\left(\left(f_{0}{-f}_{\infty }\right)e^{{-k}_{f}t}{+f}_{\infty }\right)$$


Equation (4) expresses the number of cells generated by each original cell after time *t*.

### 2.11. Animal Models of HCC

#### 2.11.1. Xenograft Mouse Model

As we previously reported [44], HuH-7 cells were used to generate a subcutaneous xenograft mouse model of HCC in SCID mice (Envigo, Milan, Italy). All in vivo experiments were performed according to the guidelines of the EU Directive (2010/63/EU) and the approval of the Veterinary Administration of the Ministry of Agriculture, Forestry and Food of the Republic of Slovenia (approval no. U34401-33/2015/5). Two types of in vivo experiments were performed, namely, long-term experiments to evaluate the effect of 5-Aza on tumor growth and survival and short-term experiments to evaluate phenotypic and molecular effects. When the largest diameter of the tumors reached 6 mm (approximately 40 mm^3^), the mice were randomly divided into different groups and subjected to specific treatments. The mice were humanely killed by CO_2_ when the tumor volume reached 300 mm^3^ (long-term experiment: 6 animals treated with 0.5 μg 5-Aza, 6 animals treated with 0.25 μg 5-Aza, and 6 animals treated with 0.9% NaCl) or three days after the first injection (short-term experiment: 6 animals treated with 0.5 μg 5-Aza and 6 animals treated with 0.9% NaCl). In the long-term experiment, 5-Aza was administered on days 0, 1, 2, 6, 7, and 8; in the short-term experiment, 5-Aza was administered on days 0, 1, and 2. In both cases, repeated administrations were performed to mimic the in vitro administration (protocol 1, Appendix A). Considering an initial tumor volume of approximately 40 mm^3^, the dose of 0.5 μg 5-Aza corresponds to an estimated initial intra-tumor concentration of 40 μM, which is very close to the most effective dose (30 μM) used in vitro. 

The tissue differentiation score was obtained considering: (1) hepatocyte nucleus/cytoplasm ratio, (2) tissue pleomorphism, i.e., the ability of hepatocytes to arrange in cords (as in normal liver tissue), (3) the amount and organization of fibrous tissue in the stroma and the presence of pseudo-lobule formation, and (4) extent of tumor necrosis.

#### 2.11.2. Xenograft Zebrafish Model

The xenograft zebrafish model of HCC was generated as previously reported [50]. JHH-6 cells were stained with 2 μg/mL FAST DiI (Sigma-Aldrich, St. Louis, MO, USA) for easy microscopic identification after microinjection (FemtoJet^®^ Eppendorf electronic microinjectors, Eppendrof, Milan, Italy) into larvae. The optimal time point for microinjection of the HCC cell line JHH-6 (500 JHH-6/embryos) into the yolk sac of zebrafish embryos was found to be two days post–egg fertilization (2 dpf). After microinjection, the growth of tumor mass was followed up to five days post-fertilization (5 dpf), which corresponds to three days post-microinjection (3 dpi). Zebrafish embryos are nearly transparent, so we were able to microscopically follow the growth of JHH-6 stained with a vital dye prior to microinjection.

5-Aza (6 μM) was added directly to the E3 medium, and then the zebrafish embryos were observed over time. Larval vital signs (i.e., heart rate) were checked using a stereomicroscope (Leica S8AP0). Pericardial edema was estimated using the area measurement tool of Sigma Scan Pro software (Jandel Scientific Software) to determine the cross-sectional area of the pericardium and the total area of the body. Results obtained from at least 30 embryos treated with 5-Aza and normalized with those from 30 untreated controls were expressed as the ratio of heart area to body area. For cartilage staining, 20 animals per group were fixed in 4% phosphate-buffered paraformaldehyde and stored in methanol at 20 °C until use. Then, staining was performed with Alcian blue solution (1% hydrochloric acid, 70% ethanol, and 0.1% Alcian blue). All images were taken at the same resolution and magnification, with the live fish positioned laterally. When necessary, embryos were anesthetized with a 1:100 dilution of 4 mg/mL Tricane (Sigma-Aldrich, St. Louis, MO, USA). The animal protocol was designed to minimize pain and discomfort of the larvae. Moreover, all experiments with zebrafish larvae were performed in the zebrafish laboratory (University of Trieste, Italy) according to the guidelines of ITA (Dgl 26/2014) in compliance with EU legislation (2010/63/UE). The zebrafish experiments and protocols were approved by a committee of the Italian Ministry of Health (Cod. 04086.N.15Y). 

#### 2.11.3. Syngeneic Rat Model

The N1-S1 rat hepatoma cell line (ECACC 90011902) was used to induce HCC in vivo in the rat model (syngeneic orthotopic). A total of 12 adult Sprague Dawley rats weighing 300–350 g (Charles River) were used for this study. Six rats were inoculated with N1-S1 cells pretreated in vitro with 5-Aza (12 µM, protocol 1, Appendix A), and six rats were inoculated with untreated N1-S1. The care and treatment of the animals and all experiments were performed in accordance with the guidelines of EEC Directive 86/609 on the care and use of laboratory animals. The protocol was approved by the institutional ethics committee of the University of Pavia. Animals were housed in the animal facility of the same university in cages with food and water ad libitum in a room with a 12 h light–dark cycle at controlled temperature and humidity. Anesthesia was induced with a gas mixture of 2–5% isoflurane in 95% oxygen and maintained with an intramuscular injection of 10 mg/kg tiletamine/zolazepam (Zoletil 50/50). Rats were placed in the supine position, and the abdominal area was shaved and disinfected with betadine. The lateral lobe of the liver was exposed through a subxiphoid incision and carefully pulled out with sterile gauze. N1-S1 cells (2 × 10^6^ cells/0.1 mL suspension) were directly and slowly injected into the liver lobe of each rat using a 25-gauge needle. After injection, the needle was withdrawn, and the injection site was compressed with a cotton swab for at least one minute to prevent infusion leakage and bleeding. The liver was then moved back to its original intra-abdominal space. Finally, the abdomen was closed in two layers (peritoneum and muscle layer, followed by the skin layer). Animals were sacrificed 2 weeks after tumor implantation, and the liver was fixed in formalin for histological analysis. The presence of tumor nodules was determined by macroscopic examination of the liver section and confirmed by histological analysis.

### 2.12. Statistical Analysis

Statistical analysis was performed using GraphPad Prism 8. All data, expressed as mean ± SEM or median with interquartile range, were tested for normal distribution using the Shapiro–Wilk test. The unpaired or paired *t*-test with or without Welch correction, Mann–Whitney test, or Wilcoxon matched-pairs signed-rank test was used to calculate *p* values, as appropriate, and *p* values < 0.05 were considered statistically significant.

## 3. Results

### 3.1. Phenotypic Effects of 5-Aza In Vitro

The different protocols used for the in vitro assays are shown schematically in Appendix A (referred to as protocols 1–9).

#### 3.1.1. Cell Proliferation

We used two human HCC cell lines, HuH-7 [35] and JHH-6 [36,37], which are known to have intermediate and low levels of differentiation, as HCC models. 5-Aza treatment (protocol 1) reduced cell number (Appendix A) and viability (Appendix A) in a dose-dependent fashion. Notably, JHH-6, the less differentiated cells, were more sensitive to 5-Aza than HuH-7, for which higher doses were required to reduce cell viability to a comparable extent to JHH-6 (Appendix A). In both cell lines, the reduction in cell growth was accompanied by a decrease in cells in the S phase and an increase in cells in G1/G0 and G2/M phases (Figure 1a,b). In contrast, 5-Aza effects, also detectable in the non-tumorigenic but immortalized IHH cells [47,52], were minimal in HLC [53], introduced as controls for normal hepatocytes (Figure 1c, Appendix A). A milder effect was also morphologically observable, as HLC maintained a polygonal shape typical of healthy hepatocytes compared to JHH-6/HuH-7 and did not exhibit notable cytoplasmic granulation (Appendix A). Remarkably, none of the cell types considered showed significant signs of apoptosis or necrosis (data not shown).

#### 3.1.2. Cell Migration and Adhesion

Using scratch assays (protocol 2), we demonstrated that 5-Aza effectively inhibited the extent and rate of migration of JHH-6 (Figure 1d,f and Appendix A); this suggests that 5-Aza has the potential to decrease the formation of the intrahepatic metastasis occurring in HCC. Because JHH-6 cells were maintained at low serum concentrations and in the presence of the anti-proliferative agent mitomycin C, we can reasonably rule out a major influence of cell growth on migration. Interestingly, the time-lapse scratch test showed a maximum reduction in wound closure between 20 and 40 h after the start of the test (Appendix A). The scratch test yielded comparable results in HuH-7 (Figure 1f and Appendix A), although higher doses of 5-Aza were required to reduce cell migration to an extent comparable to JHH-6 (compare Figure 1d,e). The FATIMA assay (protocol 1) confirmed the above results (Figure 1g,h). Finally, 5-Aza treatment reduced JHH-6 adhesion in a dose-dependent manner (Appendix A). 

#### 3.1.3. Structure of the Cytoskeleton

5-Aza induced a reduction in cell proliferation/migration, both closely correlated with cytoskeletal remodeling; this prompted us to explore possible effects on β-tubulin and β-actin, relevant components of the eukaryotic cytoskeleton [54]. 5-Aza (Figure 2a,b) induced the disorganization of β-tubulin and β-actin filaments in JHH-6/HuH-7 (protocol 1) compared to untreated cells. This was accompanied by an increased ratio of polymerized/cytosolic α-tubulin, further supporting the notion of the disruption of the normal cytoskeletal structure (Figure 2c). We also observed an increase in nuclei with abnormal polylobate shapes (Figure 2d,e) and cells with altered shapes (Figure 2a,b). All of these changes suggest that 5-Aza has an important effect on the architecture of the cytoskeleton, which may influence cell growth and migration. 

### 3.2. Molecular Effects of 5-Aza In Vitro

To uncover the molecular mechanisms responsible for the phenotypic consequences of 5-Aza treatment, we examined its effects on miR expression. In JHH-6, microarray analysis (protocol 1) revealed that eight miRs were downregulated, and three were upregulated (Figure 3a). We focused our attention on miR-139-5p because of its documented involvement in HCC [14].

#### 3.2.1. miR-139-5p Is Involved in the Phenotypic Effects of 5-Aza 

The 5-Aza-induced overexpression of miR-139-5p was confirmed in JHH-6/HuH-7 by real-time PCR (Appendix A). Remarkably, the upregulation of miRNA 139-5p paralleled that of the phosphodiesterase 2A (PDE2A) gene, in which the miR-139-5p sequence is located [55] (Appendix A). Notably, miR-139-5p was not upregulated by 5-Aza in IHH, suggesting that this is a phenomenon restricted to tumorigenic HCC cells.

Transfection of miR-139-5p (for the intracellular level after transfection, see Appendix A, protocol 3) reproduced the 5-Aza effects by: (1) altering β-tubulin organization and cell/nucleus shape Appendix A), (2) decreasing cell viability/cell number (Appendix A), (3) increasing cells in G1/G0 and G2/M phases (Appendix A), and (4) decreasing JHH-6 migration in the scratch assay (due to their growth characteristics, HuH-7 cells were not suitable for this test) (Appendix A, protocol 4). Notably, antagomiR-139-5p (which antagonizes miR-139-5p, protocol 5) attenuated the effect of 5-Aza on JHH-6/HuH-7 viability (Figure 3b,c) and JHH-6 migration (due to their growth characteristics, HuH-7 cells were not suitable for this test) (Figure 3d). As for 5-Aza treatment, neither significant necrosis nor apoptosis was induced by miR-139-5p (data not shown). Thus, miR-139-5p mimicked 5-Aza effects, supporting its involvement in the cell phenotype modifications induced by 5-Aza. 

#### 3.2.2. Upregulation of miR-139-5p Reduces ROCK2 Levels

The 5-Aza-induced upregulation of miR-139-5p prompted us to investigate the involvement of ROCK2, a demonstrated target of miR-139-5p [16] that also plays a role in HCC [19,21]. 

In JHH-6/HuH-7, 5-Aza (protocol 1) induced a dose-dependent reduction in ROCK2 protein (Figure 4a,b) and mRNA levels (Figure 4c,d). Remarkably, no reduction was observed for ROCK1 (Appendix A), the other member of the ROCK family that shares 64% amino acid sequence homology with ROCK2.

A decrease in the ROCK2 protein was related to the upregulation of miR-139-5p, as its transfection (protocol 3) reduced ROCK2 protein levels (Figure 4e,f). ROCK2 mRNA was not decreased by miR-139-5p (Figure 4g), suggesting post-transcriptional regulation of ROCK2 levels, as expected on the basis of the classical miR-mediated effect. The fact that the ROCK2 mRNA level increases in JHH-6 (Figure 4g left) may be due to a “reactive” tendency of these cells to counteract the decrease in the protein by increasing transcription.

#### 3.2.3. Downregulation of ROCK2 Is Involved in the Phenotypic Effects Induced by 5-Aza

To clarify the role of ROCK2 downregulation in the anti-proliferative effects of 5-Aza, JHH-6/HuH-7 cells overexpressing ROCK2 were treated with 5-Aza (protocol 6). Compared to the control (cells overexpressing EGFP), overexpression of ROCK2 (proved at mRNA and protein levels, see Appendix A) attenuated the anti-proliferative effect of 5-Aza in JHH-6, as determined by cell counting (Figure 5a). In HuH-7, we observed a tendency toward this phenomenon. The pro-proliferative role of ROCK2 in JHH-6/HuH-7 was further demonstrated by siRNA silencing (siROCK2, Appendix A), which (protocol 3) resulted in a significant decrease in cell growth (Appendix A). Finally, the pro-proliferative role of ROCK2 was confirmed by the ROCK inhibitor Y-27632 [56] (protocol 7), which significantly decreased cell viability (Appendix A). Thus, ROCK2 contributes to the growth of JHH-6/HuH-7, and its overexpression attenuates the anti-proliferative effect of 5-Aza.

Since miR-139-5p transfection mimicked the anti-migratory effect of 5-Aza (Appendix A), we examined the contribution of ROCK2, the downstream target of miR-139-5p, to migration. The ROCK inhibitor Y-27632 (protocol 7) significantly reduced cell migration (Appendix A), suggesting that ROCK2 contributes to migration.

The above data suggest that miR-139-5p upregulation and ROCK2 downregulation are functionally involved in the anti-proliferative/anti-migratory effects of 5-Aza (see scheme in Figure 5b). 

#### 3.2.4. Involvement of CyD1 and E2F1 in 5-Aza Induction of Cell Accumulation in G1 Phase

ROCK2 promotes the transition to the G1/S phase via the upregulation of cyclin D1 [22]. Thus, given the G1/S cell accumulation induced by 5-Aza, we investigated the 5-Aza effect on cyclin D1 (protocol 1). We observed a significant decrease in cyclin D1 levels in JHH-6/HuH-7 (Figure 6a,b). This was confirmed by miR-139-5p transfection (Figure 6c,d), supporting the role of the miR-139-5p/ROCK2 pathway in the downregulation of cyclin D1. 

Cyclin D1 activates the transcription factor E2F1, a key promoter of the transition from the G1 to S phase in many cells, including HCC cells [29]. Treatment with 5-Aza (protocol 1) significantly decreased E2F1 protein (Figure 6e,f) and mRNA levels (Appendix A). E2F1 reduction was confirmed by miR-139-5p transfection (Figure 6g,h), further supporting the role of the miR-139-5p/ROCK2 pathway in E2F1 downregulation. The ROCK2/E2F1 relationship was also evidenced by the fact that ROCK2 overexpression (protocol 6 modified) increased E2F1 mRNA levels (Figure 6i). Moreover, the reduction in E2F1 is functionally involved in the anti-proliferative effect of 5-Aza, as overexpression of E2F1 attenuated the anti-proliferative effect of 5-Aza (Figure 6j). Thus, our data suggest that the accumulation of JHH-6/HuH-7 in G1 after 5-Aza treatment may be due to cyclin D1/E2F1 reduction consequent to ROCK2 downregulation induced by the upregulation of miR-139-5p (see scheme in Figure 5b). 

#### 3.2.5. Involvement of CyB1 and p27^Kip1^ in 5-Aza Induction of Cell Accumulation in G2/M phase

ROCK2 can reduce the level of the cell cycle inhibitor p27^Kip1^ [22], thereby promoting cell cycle progression. p27^Kip1^ downregulates the cell cycle with different mechanisms, including the impairment of the CDK1/cyclin B1 complex, which can no longer promote the G2/M transition [30]. Given the increase in G2/M cells observed after treatment with 5-Aza (Figure 1a,b) and miR-139-5p (Appendix A), we examined p27^Kip1^ levels. 5-Aza increased p27^Kip1^ protein levels (protocol 1) in JHH-6/HuH-7 (Figure 7a,b), as did miR-139-5p mimic transfection (Figure 7c,d, protocol 3). This supports a role for the miR-139-5p/ROCK2 pathway in the upregulation of p27^Kip1^ and thus in the observed G2/M cell accumulation (see scheme in Figure 5b). 

G2/M cell accumulation may not be solely dependent on the increased p27^Kip1^ levels, as we also observed a decrease in cyclin B1 (which promotes the G2–M transition) levels after both 5-Aza treatment (Figure 7e–h, protocol 1) and miR-139-5p transfection (Figure 7i,j, protocol 3). While the mechanism determining the decrease in cyclin B1 after 5-Aza or miR-139-5p administration deserves further investigation, our data suggest that both the downregulation of cyclin B1 and the upregulation of p27^Kip1^ may contribute to the G2/M cell accumulation observed after 5-Aza treatment (see scheme in Figure 5b). 

#### 3.2.6. Involvement of MMP-2 in the 5-Aza-Dependent Reduction in Cell Migration

ROCK2 promotes HCC migration by stabilizing MMP-2 [19], thereby promoting extracellular matrix degradation. The reduction in JHH-6/HuH-7 migration observed after both 5-Aza (Figure 1d–h) and miR-139-5p administration in JHH-6 (Appendix A) prompted us to investigate the role of MMP-2 in the anti-migratory effect of 5-Aza. 

5-Aza (protocol 1) not only significantly decreased MMP-2 protein (Figure 8a,b) and mRNA levels (Appendix A) but also downregulated MMP-2 activity, as demonstrated by zymography (Figure 8c,d) in JHH-6 (for technical reasons, zymography was not possible in HuH-7). Remarkably, MMP-2 reduction was also confirmed after miR-139-5p administration (Figure 8e,f, protocol 3). 

To demonstrate the functional importance of MMP-2 reduction for the anti-migratory effects induced by 5-Aza, JHH-6/HuH-7 cells were co-treated with 5-Aza and bortezomib [43], which can prevent proteasome-induced degradation of MMP-2 [19]. Bortezomib (protocol 8) attenuated the anti-migratory effect of 5-Aza (Figure 8g,h) compared to 5-Aza treatment alone. The functional role of MMP-2 in migration was further supported by the observation that the MMP-2 inhibitor SB-3CT (protocol 9) effectively downregulated cell migration (Appendix A). Taken together, our data support the notion that the anti-migratory effect of 5-Aza is related to the upregulation of miR-139-5p, which leads to the downregulation of ROCK2 and, consequently, MMP-2 (see scheme in Figure 5b). 

### 3.3. Molecular Effects of 5-Aza In Vivo

HuH-7 cells were used to generate a subcutaneous xenograft mouse model, as HuH7 can generate tumors in mice. JHH-6 cells, which cannot graft into mice [57,58], were used to generate a xenograft zebrafish model of HCC [50]. Finally, the rat HCC cell line N1-S1 was used to generate an orthotopic syngeneic rat model of HCC [38]. 

#### 3.3.1. Xenograft Mouse Model of HCC

In the HuH-7 subcutaneous xenograft mouse model of HCC [44], we injected 0.5 μg of 5-Aza into the tumor in two rounds of three consecutive administrations (long-term experiment, Figure 9a left). This amount of 5-Aza (corresponding to an estimated initial intra-tumor concentration of 40 μM) was chosen because it is very similar to the most effective concentration used in vitro for HuH-7 (30 μM). Compared to control animals, 5-Aza was able to significantly delay tumor growth (Figure 9a left). The effect was dose-dependent, as injection of 0.25 μg of 5-Aza resulted in a more limited delay of tumor growth but still significantly reduced tumor growth. To gain deeper insight into the effect of 5-Aza on tumor growth, we used a simple but powerful mathematical model that we developed. Equation (3) (see Materials and Methods and Appendix A) was fitted to the data shown in Figure 9a (left). This procedure allowed us to determine the increase in tumor volume using VD +, which expresses the number of cells generated from each original cell after time t (Figure 9a right). While ~1.2 cells were generated from each original tumor cell at day 9 in control animals, this number decreased to 0.56 and 0.25 in animals treated with 0.25 and 0.5 μg of 5-Aza, respectively. Finally, 5-Aza (0.5 μg) was also able to prolong the survival of animals compared to control animals (Figure 9b). 

To explore the mechanisms of action of 5-Aza in vivo, we performed a short-term experiment limited to the fourth day after a round of three consecutive administrations (Appendix A) to mimic the in vitro protocol. The tumor growth delay was similar to that observed in the long-term experiment for the same period. Moreover, we confirmed the in vitro molecular findings, i.e., the increase in miR-139-5p levels, the decrease in ROCK2/cyclin D1/E2F1/cyclin B1/MMP-2 levels (Figure 9c–i), and the tendency of p27^Kip1^ to increase (Figure 9j). Finally, histological analysis of tumor tissues revealed the presence of polygonal structures reminiscent of liver lobules, suggesting that a differentiation effect is induced by 5-Aza (Appendix A). Moreover, the differentiation score was significantly higher in tumors from mice treated with 5-Aza than in tumors from the control group (Appendix A). This was consistent with the increased expression of albumin, a known marker of liver differentiation [59], in animals treated with 5-Aza (Appendix A). 

#### 3.3.2. Xenograft Zebrafish Model of HCC

The HCC zebrafish model that we developed [50] was used to study the in vivo effects of 5-Aza in JHH-6. After the addition of 5-Aza (6 μM) to the zebrafish E3 medium, we observed a significant reduction in JHH-6 growth compared to the non-treated larvae (Figure 10a and Appendix A). Consistent with this, we also observed a reduction in the expression of the human proliferation marker Ki67 (Figure 10b). Moreover, larvae survival was improved (Figure 10c). Although we attempted to determine the levels of miR-139-5p and the other molecular targets examined in vitro, the overabundant zebrafish RNA impeded this evaluation. Due to the small amount of human protein in the tumor mass, protein quantification was not possible.

Using the Tg(fli1:EGFP) zebrafish strain, which expresses enhanced green fluorescent protein (EGFP) [60] throughout the vasculature, we followed the effects of 5-Aza on tumor angiogenesis. We analyzed the formation of branched sub-intestinal vessel plexuses (SIVPs), which are induced by the tumor mass via the production of proangiogenic factors [61]. We observed a decrease in branched SIVPs in 5-Aza-treated larvae compared to non-treated larvae (Appendix A). Consistent with this, we detected a parallel decrease in the expression of the proangiogenic factors VEGFA, FLK-1, and CXCL8 in 5-Aza-treated (+JHH-6) compared to untreated (+JHH-6) larvae (Figure 10d). 

Finally, we investigated the possible non-specific toxic effects of 5-Aza on larvae. We considered three main parameters typically used to assess toxicity in this animal model: pericardium/body area ratio, heart rate, and development of the bone-cartilage system [62,63]. None of the three parameters differed significantly between 5-Aza-treated and non-treated animals (Appendix A) injected with JHH-6. 

#### 3.3.3. Orthotopic Autologous Rat Model of HCC

For the final evaluation of the efficacy of 5-Aza in vivo, we used an orthotopic syngeneic rat model of HCC based on the use of the N1-S1 rat cell line [38]. As with JHH-6/HuH-7, 5-Aza was effective in reducing the growth and viability of N1-S1 cells in vitro as a function of dose and time (Appendix A, protocol 1). In addition, as with JHH-6/HuH-7, we observed an increase in G1/G0 and a decrease in S-phase cell numbers (Appendix A). These data demonstrate the efficacy of 5-Aza in N1-S1.

To generate the orthotopic syngeneic HCC model, we used a sub-Glissonean inoculation of N1-S1 preceded by surgical operation of the animals (Appendix A). The inoculated N1-S1 formed white nodules after 15 days, which were readily visible on inspection and histological examination (Appendix A). To test the efficacy of 5-Aza in preventing the colonization of cancer cells, we inoculated N1-S1 cells that were either pretreated in vitro with 5-Aza (12 μM, protocol 1) or untreated. Fifteen days after inoculation, animals were sacrificed to assess tumor development. While we did not detect nodules (macroscopically/histologically) in animals inoculated with N1-S1 pretreated with 5-Aza, the absence of pretreatment resulted in nodule formation in 71% of animals.

## 4. Discussion

The aberrant DNA methylation detected in HCC [10] suggests that demethylating drugs such as 5-Aza may potentially be of therapeutic value [64]. However, most 5-Aza mechanisms of action in HCC are still unknown. Here, we elucidate some of the molecular effects of5-Aza in multiple in vitro and in vivo HCC models, thus minimizing the possibility of model-dependent artifacts. 

### 4.1. miR-139-5p and ROCK2

We show for the first time that 5-Aza can counteract the development of HCC via upregulation of miR-139-*p* and downregulation of ROCK2. This is not a trivial aspect, as miR-139-5p and ROCK2 play important roles in HCC [14,20,21]. Our data demonstrate that both the upregulation of miR-139-5p and the downregulation of ROCK2 are functionally involved in the 5-Aza effects, as: (1) miR-139-5p transfection can recapitulate the 5-Aza effects (Appendix A, Figure 4e–g), (2) the miR-139-5p antagomir attenuates the 5-Aza effects (Figure 3b–d), and (3) ROCK2 overexpression reduces 5-Aza effects (Figure 5a). 

The demonstration that 5-Aza (Figure 4c,d) but not miR-139-5p (Figure 4g) reduces ROCK2 mRNA levels suggests that miR-139-5p acts as a true miR and that 5-Aza affects both transcription (unknown mechanism) and translation (via miR-139-5 upregulation) of ROCK2. This observation could explain the more pronounced reduction in ROCK2 protein induced by 5-Aza compared to miR-139-5p alone (compare Figure 4a,b with Figure 4e,f). We can rule out that this difference is due to a suboptimal intracellular concentration of miR-139-5p, as miR-139-5p levels increased 100-fold after transfection (Appendix A). 

Our data show that 5-Aza downregulates ROCK2 (Figure 4a–d), but not ROCK1 (Appendix A), the other member of the family. Thus, in the HCC cell models studied, targeting ROCK2 per se is sufficient to downregulate cell growth and migration, as confirmed by miR-139-5p (Appendix A), by siRNA targeting of ROCK2 (Appendix A), and by chemical inhibition of ROCK2 (Appendix A). Previous studies in HCC [21,65] and in other human cancers are consistent with our observations [66,67]. However, this aspect may be somewhat tissue-dependent, as, for example, in non-small cell lung cancer and melanoma, only targeting both ROCK1/ROCK2 was able to prevent tumor growth [68]. 

### 4.2. ROCK2 and Cyclin D1/E2F1/p27 ^kip^/Cyclin B1

We demonstrated that the reduction in ROCK2 levels induced by 5-Aza and miR-139-5p was accompanied by a marked reduction in the growth of HCC cells (Figure 1a,b and Figure 9a, Appendix A). This phenomenon may have contributed to the disrupted actin/tubulin filament organization observed in cells treated with 5-Aza (Figure 2a,b). This hypothesis is based on the fact that ROCK2 requires an intact actin cytoskeleton to promote cell proliferation [22]. Moreover, alteration of the actin cytoskeleton leads to the impairment of the ROCK2-mediated upregulation of cyclin D1 [22]. Consistent with this, our data show a reduction in cyclin D1 (Figure 6a–d), supporting the notion that 5-Aza in HCC cells may also downregulate cell growth via cytoskeleton disorganization, which in turn prevents ROCK2-mediated upregulation of cyclin D1.

We also show that impairment of ROCK2 leads to a decrease in E2F1 levels, a promoter of the transition from the G1 to S phase. Since cyclin D1 activates the transcription factor E2F1 [29], it is possible that the decrease in E2F1 depends on the 5-Aza-mediated downregulation of cyclin D1. The fact that E2F1 drives its own transcription can explain the observed reduction in E2F1 mRNA (Appendix A; Figure 9g). The indirect regulation of E2F1 transcription by ROCK2 is also supported by evidence that when ROCK2 was overexpressed, E2F1 mRNA increased significantly (Figure 6i). Since overexpression of E2F1 attenuated the anti-proliferative effect of 5-Aza (Figure 6j), we demonstrated that downregulation of E2F1 is functionally involved in the anti-proliferative effect of 5-Aza. Thus, we propose for the first time that a link exists between ROCK2 and E2F1 in HCC. Notably, the observed downregulation of cyclin D1/E2F1 supports G0/G1 cell accumulation (Figure 1a,b), to which the upregulation of the cell cycle inhibitor p27^kip1^ contributes (Figure 7a–d). It is known that p27^kip1^ impairs the CDK4-6/cyclin D1 [30] complex, which is required for G1 phase completion. The fact that 5-Aza downregulates cyclin D1 and E2F1, both of which are involved in the pathogenesis of HCC [24,25,28,29], supports the rationale for using 5-Aza in HCC.

Different mechanisms contribute to explaining the observed G2/M accumulation (Figure 1a,b; Appendix A). First, p27^kip1^ upregulation may have impaired the CDK1/cyclin B1 complex, which is required for the G2/M transition [30]. Second, the downregulation of cyclin B1 levels (Figure 7f–j), a known promoter of the G2/M transition, may have contributed to the phenomenon. While the mechanism responsible for the downregulation of cyclin B1 by 5-Aza requires further investigation, our data suggest that transcriptional impairment is involved in this phenomenon (Figure 7g,h; Appendix A). Third, the altered polymerized/cytosolic α-tubulin ratio (Figure 2c) may take part in the G2/M block, as the G2/M transition requires the correct dynamics of α- and β-tubulin [69].

Taken together, our data provide clues to the molecular mechanism (Figure 5b) that helps explain the anti-proliferative effect of 5-Aza in HCC. 

### 4.3. ROCK2 and MMP-2

Our data show that by downregulating ROCK2 via the upregulation of miR-139-5p, 5-Aza reduces the level and functionality of MMP-2 (Figure 8a–f and Figure 9i) and thus impairs HCC cell migration (Figure 1d–h). Based on previous observations [19], it can be speculated that the ROCK2 decrease leads to accelerated degradation of MMP-2 via the proteasome, which in turn results in a decrease in MMP-2. Our data support this hypothesis, as treatment of HCC cells with the proteasome inhibitor bortezomib significantly attenuated the anti-migratory effect of 5-Aza (Figure 8g,h). Remarkably, bortezomib alone did not improve migratory ability, probably because no further enhancement of the migration-promoting effect occurs above a certain MMP-2 level in our cellular HCC models (Figure 8g,h: compare NTC with/without bortezomib treatment). 

Destabilization of the MMP-2 protein by decreasing ROCK2 is not the only reason for the decrease in MMP-2, as 5-Aza also downregulates MMP-2 mRNA (Appendix A). The quantitative contribution of these two mechanisms to the MMP-2 decline remains to be elucidated. Nevertheless, the known involvement of MMP-2 in HCC cell migration [19] underscores the rationale for using 5-Aza in HCC. 

### 4.4. Animal Models

In the xenograft mouse model of HCC, 5-Aza was able to recapitulate the phenotypic effects observed in vitro with HuH-7 (Figure 9a,b; Appendix A). With respect to tumor growth, our mathematical analysis (Figure 9a right and left) allowed us to determine the proliferation rate, which was remarkably reduced in animals treated with 0.5 μg of 5-Aza compared to the control (0.25 versus 1.2). Without this analysis, the effects of 5-Aza could only have been roughly inferred from the growth curves. Thus, our mathematical model provides a powerful tool to accurately and objectively compare the efficacy of different anti-tumor approaches (different anti-tumor drugs, dosages, delivery systems, etc.) based on the tumor growth curve.

At the histological level, our data show some degree of tumor differentiation induced by 5-Aza (Appendix A), which was accompanied by an increase in mRNA levels of the differentiation marker albumin (Appendix A). This observation warrants further investigation but is consistent with previous data [34,70] and supports the rationale for using 5-Aza in HCC. 

At the molecular level, we confirmed the observations made in vitro in HuH-7 (Figure 9c–i). For p27^kip1^ (Figure 9j), we only observed a tendency towards upregulation. This might be due to the presence of mouse cells (vascular cells, white blood cells, etc.) in the xenografted tumor mass, where the effect of 5-Aza on p27^kip1^ levels is unknown. In addition, the upregulation of p27^kip1^ in vitro (about +20%, Figure 7b) was significant but not extreme. This may have favored the confounding effect of mouse cells present in tumor tissue. Consistent with this possibility, for targets such as ROCK2, cyclin D1, E2F1, and cyclin B1, where the variation in expression was more pronounced in HuH-7 in vitro (60%, 50%, 70%, and 80%, respectively, Figure 4b, Figure 6b,f and Figure 7f), a significant difference was clearly detectable in vivo as well. 

Although 5-Aza was administered locally in the mouse model, it had no significant systemic toxicity, as determined by monitoring animal weight and other parameters, such as posture, condition of skin and coat, locomotion, respiration, and food intake. In addition to the agreement with previous data [34] and the minor cytotoxicity in vitro in non-tumor cells (Appendix A), we were able to confirm these results in the zebrafish model. Here, there was no evidence of relevant toxicity in the assessment of pericardial edema, heart rate, or organization of the bone-cartilage system (Appendix A). It should be noted that 5-Aza can be considered systemically administered in zebrafish, as it was added directly to the E3 medium.

In the zebrafish model, as in the mouse model, we observed a reduction in tumor growth, an increased larval survival rate (Figure 10a,c), and a reduction in tumor angiogenesis (Appendix A). Of note, no significant differences in proangiogenic factor expression were observed between 5-Aza-treated and non-treated animals in larvae injected with the medium alone (no JHH-6) (Figure 10d). This suggests that 5-Aza has no significant effect on angiogenesis per se. Thus, the reduced angiogenesis in larvae injected with JHH-6 can be attributed to the reduced growth of JHH-6 due to 5-Aza. This observation is of particular importance considering that HCC is a highly vascularized tumor [71], and therefore, anti-angiogenic molecules are of potential benefit.

Finally, in an orthotopic syngeneic rat model of HCC, we tested the effect of 5-Aza on a third HCC cell line (N1-S1) that responded similarly to 5-Aza in vitro to JHH-6/HuH-7 (Appendix A). The main goal of this model is to demonstrate that 5-Aza is effective in reducing HCC cell grafting at the correct anatomical site and to confirm the observed reduction in the adhesion of JHH-6 in vitro (Appendix A). Although these data are very promising, they need to be implemented with the evaluation of the effects of the systemic administration of 5-Aza (in progress). In this context, we believe [71,72] that trans-arterial tumor delivery would be the most appropriate route, possibly in combination with targeted delivery of 5-Aza, as we have recently developed for anti-HCC siRNAs [44]. 

## 5. Conclusions

In this work, we elucidate some mechanisms responsible for the anti-HCC properties of 5-Aza. We show that 5-Aza upregulates miR-139-5p, which in turn causes the downregulation of ROCK2/cyclin D1/E2F1/cyclin B1 and the upregulation of p27^kip1^ (Figure 5b), inducing the accumulation of HCC cells in G1/G0 and G2/M phases of the cell cycle. In addition, downregulation of ROCK2 decreases MMP-2 levels (Figure 5b), which in turn impairs HCC cell migration. Overall, our observations contribute to elucidating the mechanisms of action of 5-Aza and deepen the knowledge about the crosstalk among ROCK2/cyclin D1/E2F1/cyclin B1/p27^kip1^/MMP-2 in HCC. Given the relatively short half-life of 5-Aza in vivo, protective and HCC-targeted delivery systems would likely be required to enhance its efficacy. Alternatively, second/third-generation demethylating molecules could be used due to their longer half-life in vivo. Despite these considerations, the data presented here strengthen the case for the potential efficacy of 5-Aza and demethylating agents in general in HCC.

## Figures and Tables

**Figure 1 cancers-14-01630-f001:**
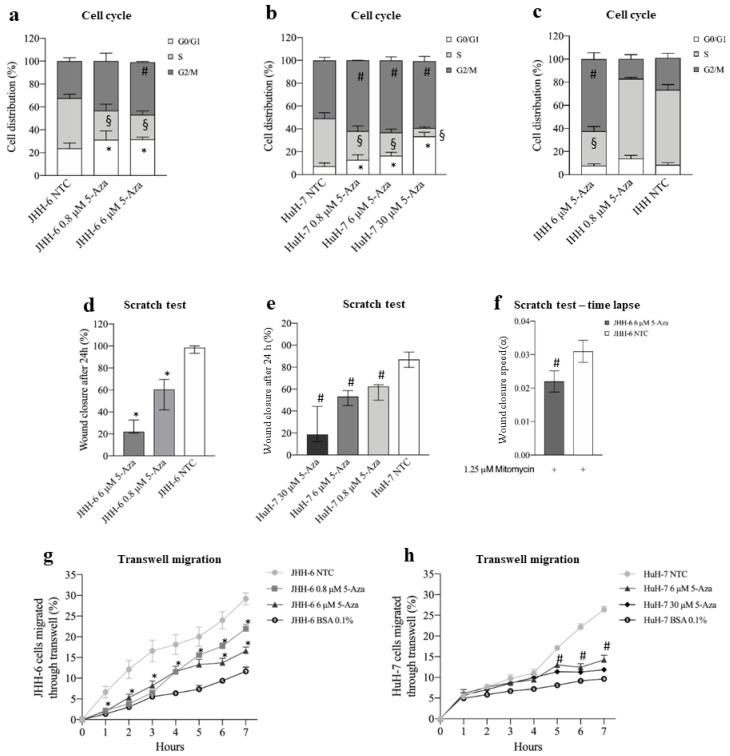
Effects of 5-Aza on cell growth and migration. (**a**–**c**) Cell cycle phase distribution. Data, expressed as mean ± SEM, are reported as percentage. JHH-6 NTC G0/G1 vs. JHH-6 0.8/6 μM 5-Aza G0/G1, * *p* < 0.021; JHH-6 NTC S vs. JHH-6 0.8/6 μM S, ^§^
*p* < 0.0087; JHH-6 NTC G2/M vs. JHH-6 0.8/6 μM G2/M, ^#^
*p* < 0.011, *n* = 5. HuH-7 NTC G0/G1 vs. HuH-7 0.8/6/30 5-Aza μM G0/G1, * *p* < 0.022; HuH-7 NTC S vs. HuH-7 0.8/6/30 5-Aza μM S, ^§^
*p* < 0.024; HuH-7 NTC G2/M vs. HuH-7 0.8/6/30 μM G2/M, ^#^
*p* < 0.045, *n* = 3. IHH NTC S vs. IHH 6 μM 5-Aza S, ^§^
*p* < 0.0003; IHH NTC G2/M vs. IHH 6 μM 5-Aza G2/M, ^#^
*p* = 0.0012, *n* = 3. (**d**,**e**) Cell migration, evaluated by scratch assay. Data, expressed as percentage, are presented as median with interquartile range; JHH-6/NTC vs. JHH-6 0.8/6 5-Aza μM, * *p* = 0.0012, *n* = 6; HuH-7/NTC vs. HuH-7 0.8/6/30 5-Aza μM, ^#^
*p* = 0.0002, *n* = 8. (**f**) Wound healing experiments were analyzed by measuring the reduction in the area of the cell-free region over time (A), normalized with respect to its initial value (A0); the slope of the linear range of the A/A0 vs. t curve can be considered a measure of the wound closure velocity α (h−1); JHH-6/NTC/mitomycin vs. JHH-6/6 μM 5-Aza/mitomycin, ^#^
*p* = 0.012, *n* = 5. (**g**,**h**) Effects on cell migration, evaluated by FATIMA assay. Data, expressed as mean ± SEM, are reported as percentage. JHH-6/NTC vs. JHH-6 0.8/6 5-Aza μM, * *p* < 0.046, *n* = 5; HuH-7/NTC vs. HuH-7 6/30 5-Aza μM, ^#^
*p* < 0.014, *n* = 4; BSA: bovine serum albumin (control-treated cells).

**Figure 2 cancers-14-01630-f002:**
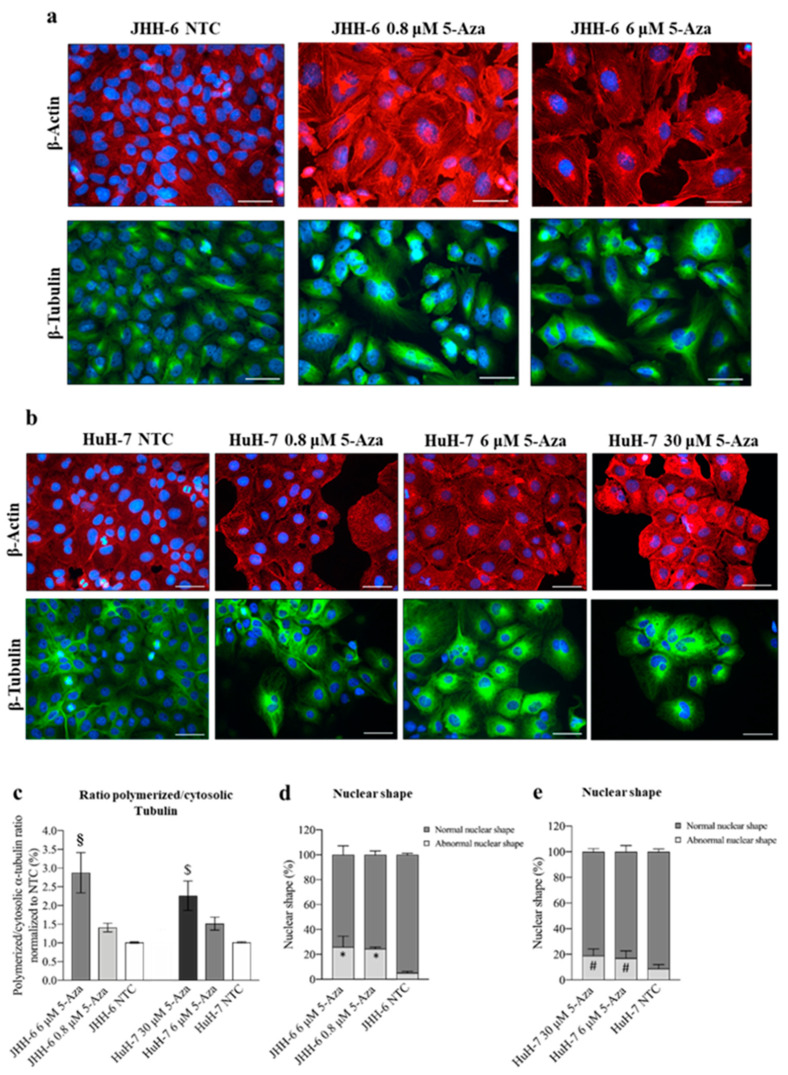
Effects of 5-Aza on cell cytoskeleton. (**a**,**b**) Immunostaining of β-actin and β-tubulin in JHH-6 and HuH-7 cells, respectively. Images were acquired with a Leica DM 2000 microscope. Red = β-actin; green = β-tubulin; blue = DAPI. Magnification 40X (bar = 100 μm). (**c**) Ratio of polymerized/cytosolic α-tubulin in 5-Aza-treated vs. non-treated cells (NTC); data are expressed as mean ± SEM; JHH-6 NTC vs. JHH-6/5-Aza, ^§^
*p*= 0.04; HuH-7 NTC vs. HuH-7/5-Aza, ^$^
*p*= 0.045, *n* = 3. (**d**,**e**) Quantification of the number of altered nuclei shape in 5-Aza-treated vs. non-treated cells. Data, expressed as mean ± SEM, are reported as percentage normalized to the average of non-treated cells (NTC). Abnormal nuclei in JHH6/NTC vs. JHH6/0,8-6 μM, * *p* < 0.0001 *n* = 60; HuH7/NTC vs. HuH7/6-30 μM, ^#^
*p* < 0.001 *n* = 60.

**Figure 3 cancers-14-01630-f003:**
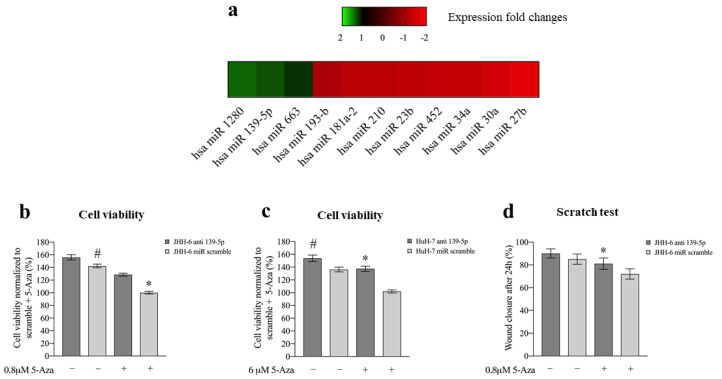
Effects of 5-Aza on miR expression and effects of miR-139-5p on cell viability and migration. (**a**) Differentially expressed miRs following 6 μM 5-Aza treatments in JHH-6 evaluated by microarray analysis. (**b**) Viability of JHH-6 treated with both antagomiR-139-5p (100 nM) and 5-Aza. Data, normalized to the average of JHH-6 scr + 5-Aza, are represented as mean ± SEM, *n* = 3; JHH-6 antagomiR-139-5p vs. JHH-6 scr, ^#^
*p* = 0.0123; JHH-6 antagomiR-139-5p + 5-Aza vs. JHH-6 scr + 5-Aza, * *p* = 0.0001. (**c**) Viability of HuH-7 treated by both antagomiR-139-5p and 5-Aza. Data, normalized to the average of HuH-7 scr + 5-Aza, are represented as mean ± SEM, *n* = 3. HuH-7 antagomiR-139-5p vs. HuH-7 scr, ^#^
*p* = 0.0065; HuH-7 antagomiR-139-5p + 5-Aza vs. HuH-7 scr + 5-Aza, * *p* = 0.0001. (**d**) Migration of JHH-6 treated with both antagomiR-139-5p and 5-Aza. JHH-6 antagomiR-139-5p + 5-Aza vs. JHH-6 miR scr + 5-Aza, * *p* = 0.002. Data are presented as mean ± SEM, *n* = 3.

**Figure 4 cancers-14-01630-f004:**
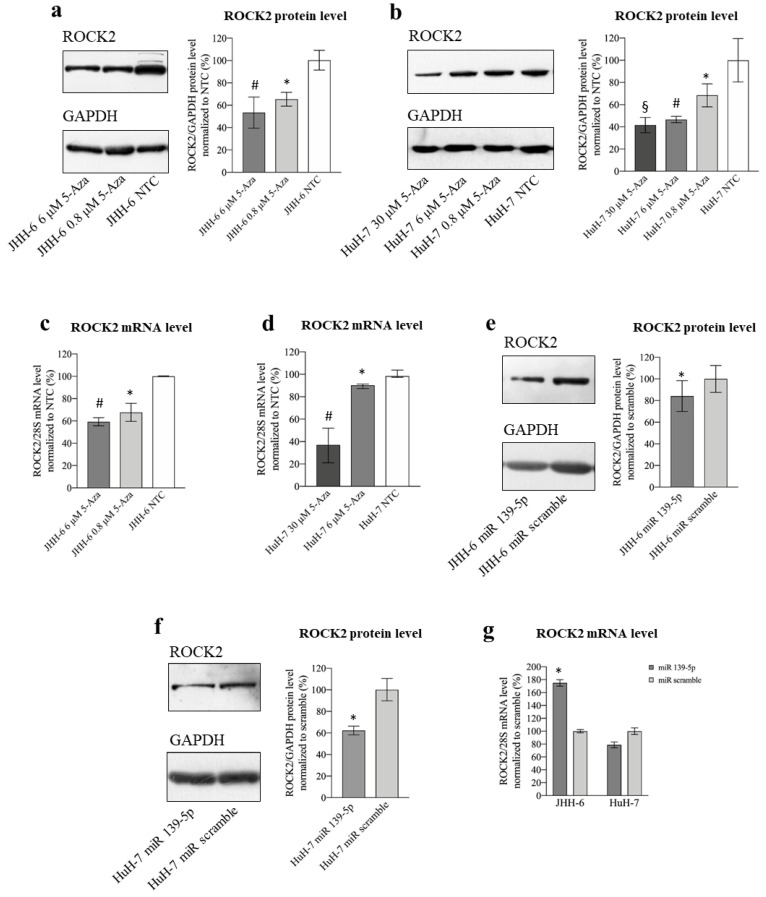
*Effects of 5-Aza and miR-139-5p on ROCK2 expression.* (**a**) Effects of 5-Aza on ROCK2 protein level in JHH-6. Left, representative Western blot (GAPDH was introduced for normalization); right, data normalized to GAPDH levels and to the average of non-treated cells (NTC) are reported in % as mean ± SEM, *n* = 6; NTC vs. 0.8 μM, * *p* = 0.0398; NTC vs. 6 μM, ^#^
*p* = 0.0177. (**b**) Effects of 5-Aza on ROCK2 protein level in HuH-7. Left, representative Western blot; right, data normalized to GAPDH levels and to the average of NTC are reported in % as mean ± SEM, *n* = 4; NTC vs. 0.8 μM, * *p* = 0.0135; NTC vs. 6 μM, ^#^
*p* = 0.0081; NTC vs. 30 μM, ^§^
*p* = 0.038. (**c**) Effects of 5-Aza on ROCK2 mRNA level in JHH-6. Data normalized to 28S levels and to the average of NTC are reported in % as mean ± SEM, *n* = 6; NTC vs. 0.8 μM, * *p* = 0.0025; NTC vs. 6 μM, ^#^
*p* = 0.0001. (**d**) Effects of 5-Aza on ROCK2 mRNA level in HuH-7. Data normalized to 28S levels and to the average of NTC are reported in % as median with interquartile range, *n* = 6; NTC vs. 6 μM, * *p* = 0.0022; NTC vs. 30 μM, ^#^
*p* = 0.002. (**e**) Effects of miR-139-5p on ROCK2 protein level in JHH-6. Left, representative Western blot; right, data normalized to GAPDH levels and to the average of scr (scrambled miR treated cells) are reported in % as mean ± SEM, *n* = 5; scr vs. miR-139-5p, * *p* = 0.0071. (**f**) Effects of miR-139-5p on ROCK2 protein level in HuH-7. Left, representative Western blot; right, data normalized to GAPDH levels and to the average of scr are reported in % as mean ± SEM, *n* = 5; scr vs. miR-139-5p, * *p* = 0.0268. (**g**) Effects of miR-139-5p on ROCK2 mRNA level in JHH-6 and HuH-7. Data normalized to 28S levels and to the average of scr are reported in % as mean ± SEM, *n* = 3; scr vs. miR-139-5p, * *p* = 0.04.

**Figure 5 cancers-14-01630-f005:**
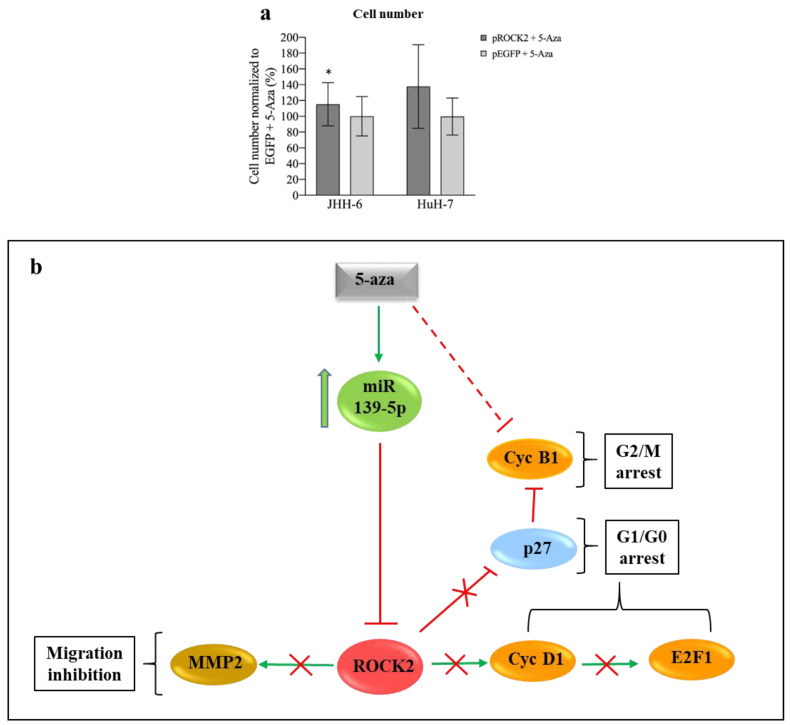
*Phenotypic effects of ROCK2 overexpression/downregulation and miR-139-5p/ROCK2 pathway.* (**a**) ROCK2 overexpression and 5-Aza treatment. Left: data in %, normalized to the average of JHH-6 overexpressing EGFP (enhanced green fluorescent protein, control cells) + 5-Aza treatment (0.8 μM), are expressed as mean ± SEM, *n* = 4; JHH-6 overexpressing ROCK2 + 5-Aza vs. JHH-6 overexpressing EGFP + 5-Aza, * *p* = 0.035. Right: HuH-7 overexpressing ROCK2 + 5-Aza vs. HuH-7 overexpressing EGFP + 5-Aza (6 μM); data in %, normalized to the average of HuH-7 overexpressing EGFP + 5-Aza, are expressed as mean ± SEM, *n* = 3. (**b**) Molecular pathways regulated by miR-139-5p and effects on cell migration and proliferation: 5-Aza upregulates miR-139-5p, which in turn leads to the downregulation of ROCK2/cyclin D1/E2F1/cyclin B1/MMP-2 and the upregulation of p27^kip^1; this results in G1/G0-G2/M cell cycle arrest and in the impairment of cell migration.

**Figure 6 cancers-14-01630-f006:**
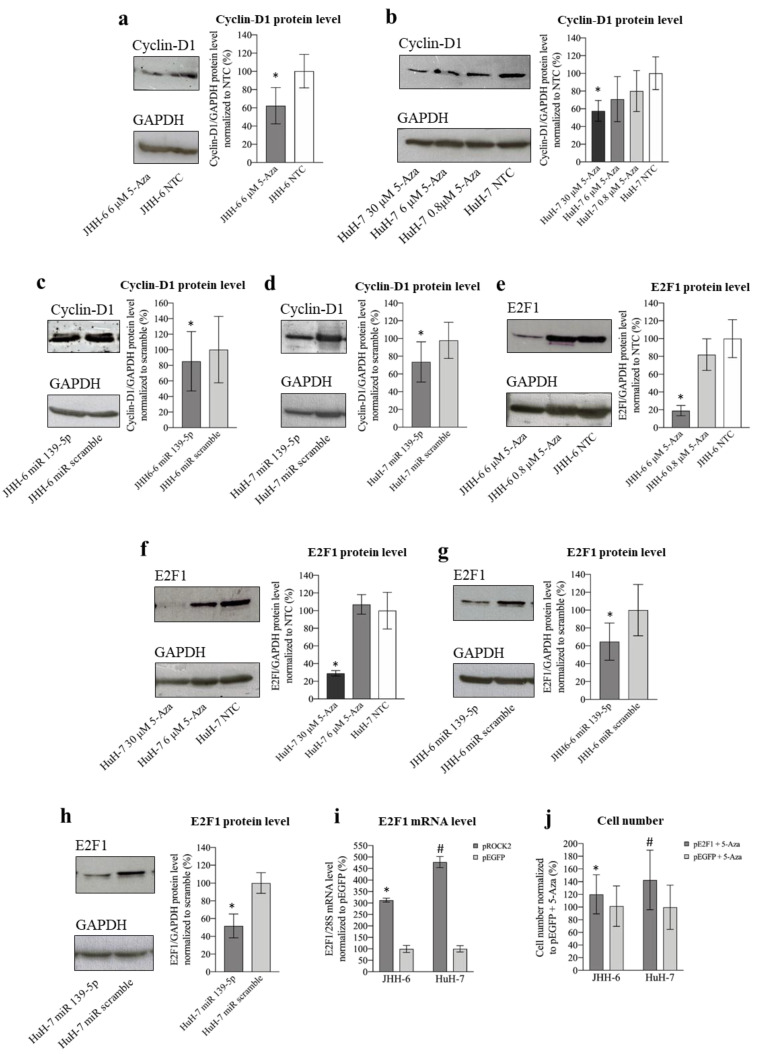
Effects of 5-Aza and miR-139-5p on cyclin D1 and E2F1 expression. (**a**) Effects of 5-Aza on cyclin D1 protein level in JHH-6. Left, representative Western blot (GAPDH was introduced for normalization); right, data normalized to GAPDH levels and to the average of non-treated cells (NTC) are reported in % as mean ± SEM, *n* = 5; NTC vs./6 μM, * *p* = 0.0005. (**b**) Effects of 5-Aza on cyclin D1 protein level in HuH-7. Left, representative Western blot; right, data normalized to GAPDH levels and to the average of NTC are reported in % as mean ± SEM, *n* = 6; NTC vs. 30 μM, * *p* = 0.048. (**c**) Effects of miR-139-5p on cyclin D1 protein level in JHH-6. Left, representative Western blot; right, data normalized to GAPDH levels and to the average of scr (scrambled miR treated cells) are reported in % as mean ± SEM, *n* = 4; scr vs. miR-139-5p-treated cells, * *p* = 0.048. (**d**) Effects of miR-139-5p on cyclin D1 protein level in HuH-7. Left, representative Western blot; right, data normalized to GAPDH levels and to the average of scr are reported in % as mean ± SEM, *n* = 5; scr vs. miR-139-5p-treated cells, * *p* = 0.045. (**e**) Effects of 5-Aza on E2F1 protein level in JHH-6. Left, representative Western blot; right, data normalized to GAPDH levels and to the average of NTC are reported in % as mean ± SEM, *n* = 3; NTC vs. 6 μM, * *p* = 0.048. (**f**) Effects of 5-Aza on E2F1 protein level in HuH-7. Left, representative Western blot; right, data normalized to GAPDH levels and to the average of NTC are reported in % as mean ± SEM, *n* = 3; NTC vs. 30 μM, * *p* = 0.044. (**g**) Effects of miR-139-5p on E2F1 protein level in JHH-6. Left, representative Western blot; right, data normalized to GAPDH levels and to the average of scr are reported in % as mean ± SEM, *n* = 3; scr vs. miR-139-5p-treated cells, * *p* = 0.04. (**h**) Effects of miR-139-5p on E2F1 protein level in HuH-7. Left, representative Western blot; right, data normalized to GAPDH levels and to the average of scr are reported in % as mean ± SEM, *n* = 3; scr vs. miR-139-5p-treated cells, * *p* = 0.015. (**i**) E2F1 mRNA levels following ROCK2 overexpression. Left JHH-6, data normalized to 28S levels and to the average of pEGFP (control plasmid, cells overexpressing EGFP) are reported in % as mean ± SEM, *n* = 4; pEGFP vs. pROCK2 (cells overexpressing ROCK2), * *p* = 0.028. Right: HuH-7, data normalized to 28S levels and to the average of pEGFP are reported in % as mean ± SEM, *n* = 4; pEGFP vs. pROCK2, ^#^
*p* = 0.028. (**j**) Effects of E2F1 overexpression on the number of 5-Aza-treated cells. Left: data in %, normalized to the average of JHH-6 overexpressing EGFP + 5-Aza treatment, are expressed as mean ± SEM, *n* = 4; JHH-6 overexpressing E2F1 + 5-Aza vs. JHH-6 overexpressing EGFP + 5-Aza, * *p* = 0.047. Right: data in %, normalized to the average of HuH-7 overexpressing EGFP + 5-Aza, are expressed as mean ± SEM, *n* = 4; HuH-7 overexpressing E2F1 + 5-Aza vs. HuH-7 overexpressing EGFP + 5-Aza, ^#^
*p* = 0.032.

**Figure 7 cancers-14-01630-f007:**
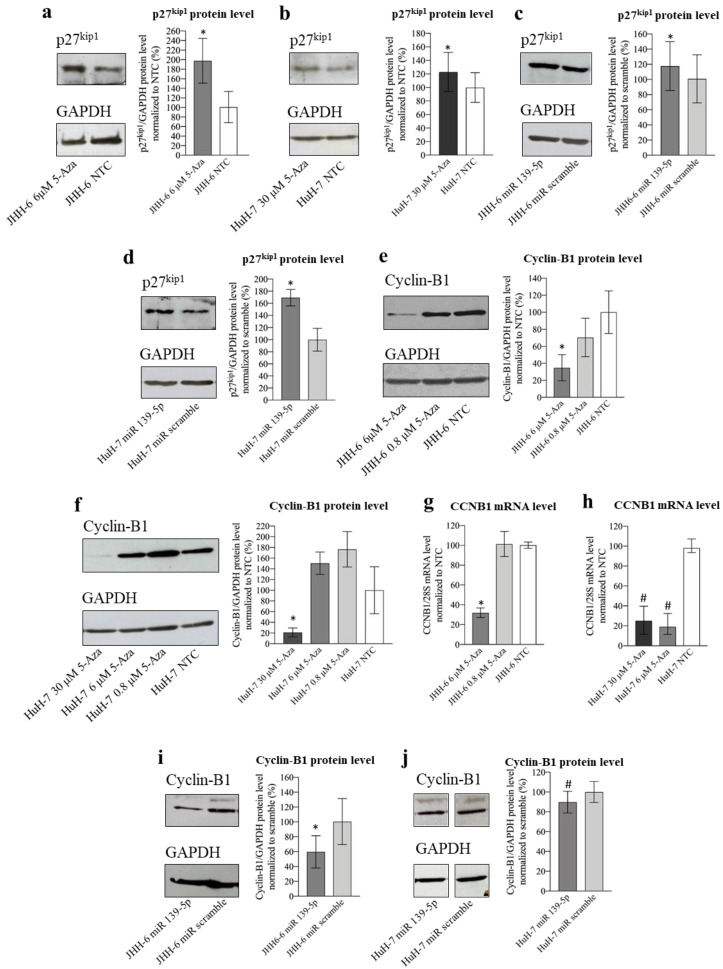
Effects of 5-Aza and miR-139-5p on p27^kip^ and cyclin B1 expression. (**a**) Effects of 5-Aza on p27^kip1^ protein level in JHH-6. Left, representative Western blot (GAPDH was introduced for normalization); right, data normalized to GAPDH levels and to the average of non-treated cells (NTC) are reported in % as mean ± SEM, *n* = 6; NTC v.s 6 μM, * *p* = 0.031. (**b**) Effects of 5-Aza on p27^kip1^ protein level in HuH-7. Left, representative Western blot; right, data normalized to GAPDH levels and to the average of NTC are reported in % as mean ± SEM, *n* = 3; NTC vs. 30 μM, * *p* = 0.018. (**c**) Effects of miR-139-5p on p27^kip1^ protein level in JHH-6. Left, representative Western blot; right, data normalized to GAPDH levels and to the average of scr (scrambled miR treated cells) are reported in % as mean ± SEM, *n* = 5; scr vs. miR-139-5p-treated cells, * *p* = 0.01. (**d**) Effects of miR-139-5p on p27^kip1^ protein level in HuH-7. Left, representative Western blot; right, data normalized to GAPDH levels and to the average of scr are reported in % as mean ± SEM, *n* = 3; scr vs. miR-139-5p-treated cells, * *p* = 0.025. (**e**) Effects of 5-Aza on cyclin B1 protein level in JHH-6. Left, representative Western blot; right, data normalized to GAPDH levels and to the average of NTC are reported in % as mean ± SEM, *n* = 6; NTC vs. 6 μM, * *p* = 0.044. (**f**) Effects of 5-Aza on cyclin B1 protein level in HuH-7. Left, representative Western blot; right, data normalized to GAPDH levels and to the average of NTC are reported in % as mean ± SEM, *n* = 4; NTC vs. 30 μM, * *p* = 0.042. (**g**,**h**) Effects of 5-Aza on CCNB1 mRNA level in JHH-6 and HuH-7, respectively. Data normalized to 28S levels and to the average of NTC are reported in % as median with interquartile range, *n* = 6; JHH-6/NTC vs. JHH-6/6 μM 5-Aza, * *p* = 0.0001; HuH-7/NTC vs. HuH-7/6-30 μM 5-Aza, ^#^
*p* < 0.0022. (**i**,**j**) Effects of miR-139-5p on cyclin B1 protein level in JHH-6 and HuH-7, respectively. In each panel, a representative Western blot is presented on the left; right, data normalized to GAPDH levels and to the average of scr are reported in % as mean ± SEM, *n* = 4; JHH-6/scr vs. JHH-6/miR-139-5p-treated cells, * *p* = 0.042, HuH-7/scr vs. HuH-7/miR-139-5p-treated cells, ^#^
*p* = 0.0002.

**Figure 8 cancers-14-01630-f008:**
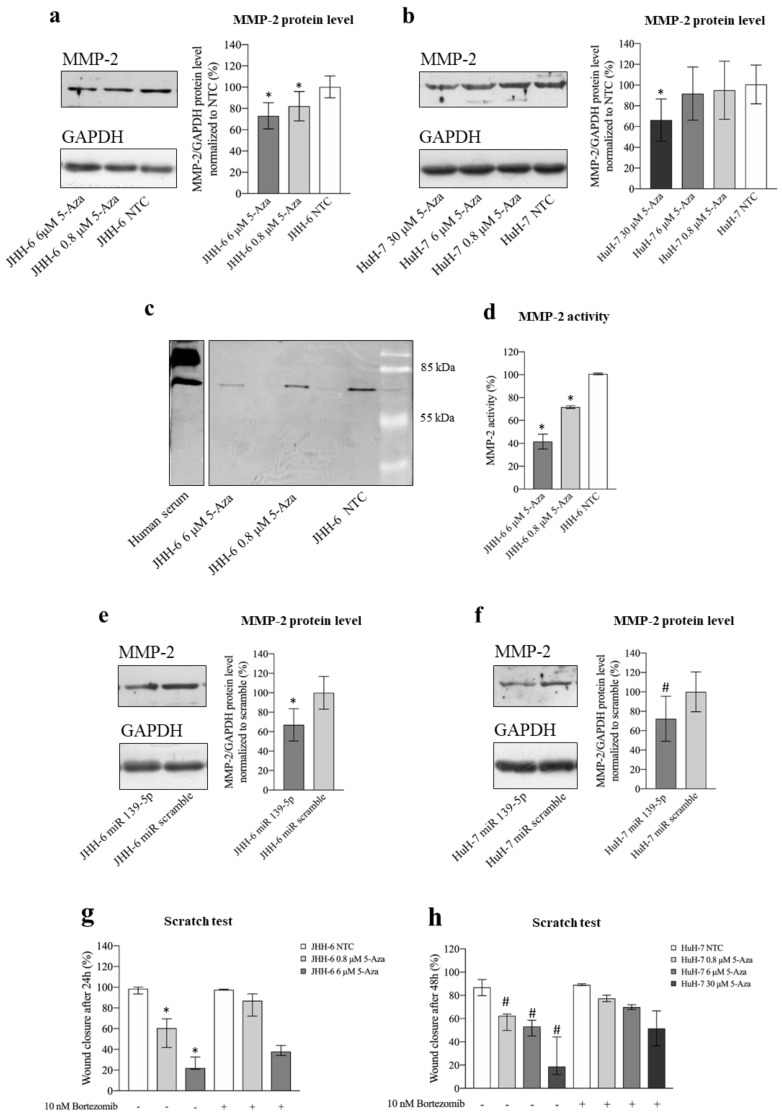
Effects of 5-Aza and miR-139-5p on MMP-2. (**a**) Effects of 5-Aza on MMP-2 protein level in JHH-6. Left, representative Western blot (GAPDH was introduced for normalization); right, data normalized to GAPDH levels and to the average of non-treated cells (NTC) are reported in % as mean ± SEM, *n* = 7; NTC vs. 0.8/6 μM, * *p* < 0.048. (**b**) Effects of 5-Aza on MMP-2 protein level in HuH-7. Left, representative Western blot; right, data normalized to GAPDH levels and to the average of NTC are reported in % as mean ± SEM, *n* = 5; NTC vs. 30 μM, * *p* = 0.016. (**c**,**d**) Evaluation of MMP-2 activity by zymography; left, representative zymogram obtained from the analysis of JHH-6 supernatants is shown; right, data are reported in % as mean ± SEM, *n* = 6; NTC vs. 0.8/6 μM, * *p* < 0.0001. (**e**,**f**) Effects of miR-139-5p on MMP-2 protein level in JHH-6 and HuH-7, respectively. In each panel, a representative Western blot is shown on the left; right, data normalized to GAPDH levels and to the average of scr (scrambled miR treated cells) are reported in % as mean ± SEM, *n* = 4; JHH-6/scr vs. JHH-6/miR-139-5p-treated cells, * *p* = 0.041; HuH-7/scr vs. HuH-7/miR-139-5p-treated cells, ^#^
*p* = 0.0031. (**g**,**h**) Scratch assay quantification in JHH-6 and HuH-7 cells treated by 5-Aza with or without the combined treatment with bortezomib. Data are represented as median with interquartile range, *n* = 4; JHH-6/0.8/6 μM aza vs. JHH-6/0.8-6 μM aza + bortezomib, * *p* < 0.0314; HuH-7/0.8-6-30 μM aza vs. HuH-7/0.8-6-30 μM aza + bortezomib, ^#^
*p* < 0.04.

**Figure 9 cancers-14-01630-f009:**
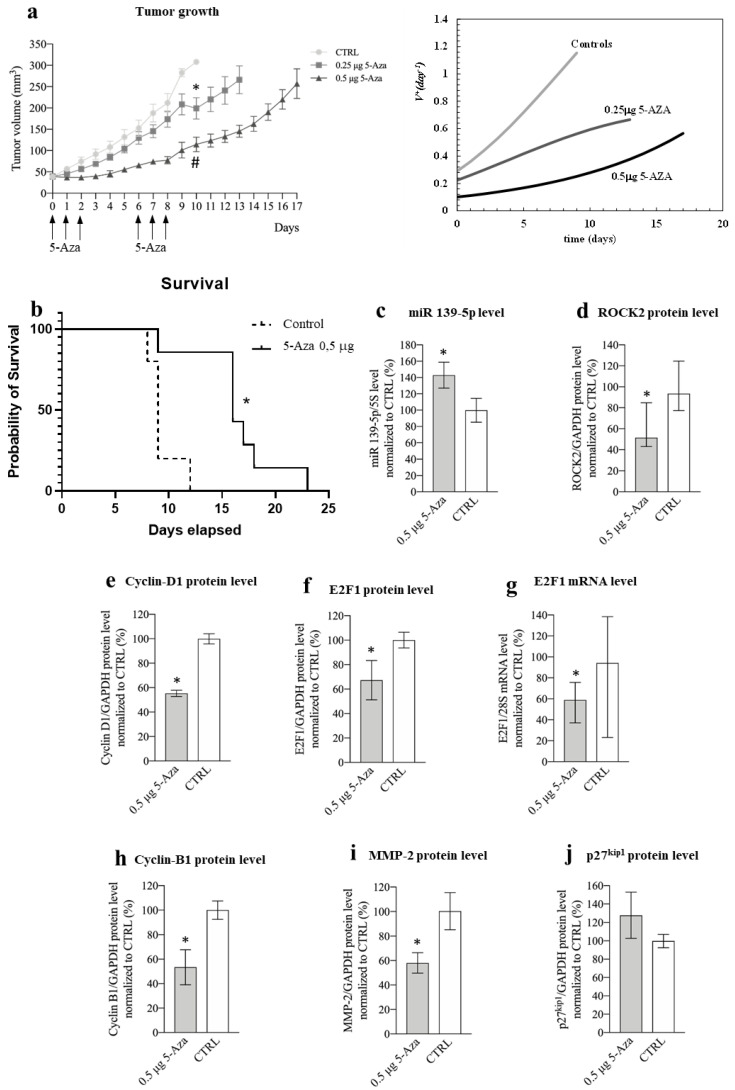
Phenotypic and molecular effects of 5-Aza in the xenograft mouse model of HCC. (**a**) Effects of 5-Aza on tumor growth. Left: the arrows indicate the administration times for 5-Aza (either 0.25 μg or 0.5 μg per injection); data are reported in mm^3^ as mean ± SEM, 0.5 μg 5-Aza/day1-10 vs. CTRL (saline solution treated animals) days 1-10, ^#^
*p* = 0.0022; 0,25μg 5-Aza/day1-10 vs. CTRL day/1-10, * *p* = 0.0036. Right: Equation (3) best fit to tumor growth data determines the dimensionless tumor speed velocity *V*_D_^+^ (Equation (4)) expressing the number of cells generated by each original cell after time. (**b**) Kaplan–Meier curve; mice were treated either with 0.5 μg 5-Aza/day or with saline solution (CTRL), log-rank (Mantel–Cox) test, * *p* = 0.03, *n* = 6. (**c**) miR-139-5p expression level; data normalized to 5s rRNA content and to the average of CTRL are reported in % as mean ± SEM, CTRL vs. 0.5 μg 5-Aza, * *p*= 0.047, *n* = 6. (**d**) Effects of 5-Aza on ROCK2 protein level; data normalized to GAPDH levels and to the average of CTRL are reported in % as median with interquartile range, *n* = 6; CTRL vs. 0.5 μg 5-Aza, * *p* = 0.041, *n* = 6. (**e**) Effects of 5-Aza on cyclin D1 protein level; data normalized to GAPDH levels and to the average of CTRL are reported in % as mean ± SEM, *n* = 6; CTRL vs. 0.5 μg 5-Aza, * *p* = 0.0001. (**f**) Effects of 5-Aza on E2F1 protein level; data normalized to GAPDH levels and to the average of CTRL are reported in % as mean ± SEM, *n* = 6; CTRL vs. 0.5 μg 5-Aza, * *p* = 0.047. (**g**) Effects of 5-Aza on E2F1 mRNA level; data normalized to 28S levels and to the average of CTRL are reported in % as median with interquartile range, CTRL vs. 0.5 μg 5-Aza, * *p* = 0.041, *n* = 6. (**h**) Effects of 5-Aza on cyclin B1 protein level; data normalized to GAPDH levels and to the average of CTRL are reported in % as mean ± SEM, *n* = 5; CTRL vs. 0.5 μg 5-Aza, * *p* = 0.002. (**i**) Effects of 5-Aza on MMP-2 protein level; data normalized to GAPDH levels and to the average of CTRL are reported in % as mean ± SEM, *n* = 6; CTRL vs. 0.5 μg 5-Aza, * *p* = 0.045. (**j**) Effects of 5-Aza on p27^kip^ protein level; data normalized to GAPDH levels and to the average of CTRL are reported in % as mean ± SEM, *n* = 5.

**Figure 10 cancers-14-01630-f010:**
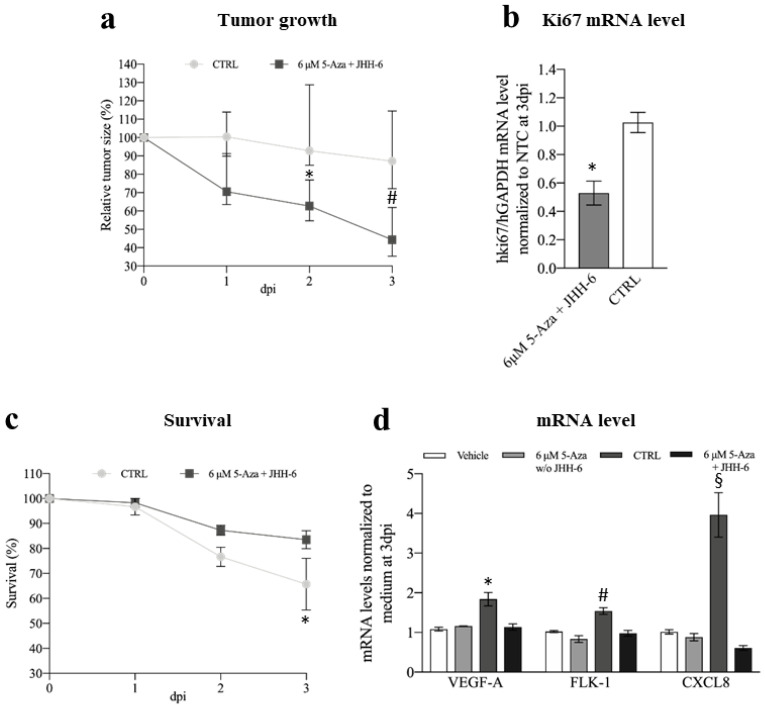
Phenotypic and molecular effects of 5-Aza in the xenograft zebrafish model of HCC. (**a**) Comparison of tumor growth between 5-Aza-treated and control zebrafish inoculated with JHH-6; data are reported in % as median with interquartile range, CTRL 2 days post-injection (dpi) vs. 6 μM 5-Aza 2dpi, * *p* = 0.025; CTRL 3 dpi vs. 6 μM 5-Aza 3dpi, # *p* = 0.0015, *n* = 45. (**b**) Effects of 5-Aza on human Ki67 mRNA level; data, normalized to GAPDH levels and to the average of CTRL, are reported as mean ± SEM, CTRL vs. 0.5 μg 5-Aza * *p* = 0.0002, *n* = 12. (**c**) Kaplan–Meier curve; zebrafish were treated either with 0.5 μg 5-Aza/day or with saline solution (CTRL), log-rank (Mantel–Cox) test, * *p* = 0.0252, *n* = 45 for each group; % of animal survival is expressed as mean ± SEM, *n* = 45. (**d**) Effects of 5-Aza on the zebrafish VEGFA, FLK-1, and CXCL8 mRNA levels (vehicle: animal not injected with JHH-6 and treated with saline; 6 μM 5-Aza *w*/*o* JHH-6: animal not injected with JHH-6 and treated with 5-Aza; CTRL: animal injected with JHH-6 and treated with saline; 6 μM 5-Aza + JHH-6: animal injected with JHH-6 and treated with 5-Aza). Data, normalized to GAPDH levels and to the average of CTRL at 3dpi, are reported as mean ± SEM. VEGFA: CTRL vs. 6 μM 5-Aza + JHH-6, * *p* = 0.0359; FLK-1: CTRL vs. 6 μM 5-Aza + JHH-6, ^#^
*p* = 0.0495; CXCL8: CTRL vs. 6 μM 5-Aza + JHH-6, ^§^
*p* < 0.0001, *n* = 4.

## Data Availability

The original data are available upon request to the corresponding author.

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
