# Peer review of "5-Azacytidine Downregulates the Proliferation and Migration of Hepatocellular Carcinoma Cells In Vitro and In Vivo by Targeting miR-139-5p/ROCK2 Pathway"

_cancers, 2022, doi:10.3390/cancers14071630_

Round 1

Reviewer 1 Report

The authors did an extensive study to understand the mechanism of the antiproliferative agent, 5-aza. The authors in particular focused on the phenotypic and molecular effects of 5-aza, and they showed in both in-vitro and in-vivo studies that upregulation of miR 139-5P is important in the anticancer activity of 5-Aza in HCC. The study design and experiments are adequate, and the results are pretty convincing. I recommend the manuscript for publication after the authors address the following points:

  1. The authors should explain the significance of the Scratch test and cell migration experiments. Are these indicators of metastasis in HCC?
  2. In line 591, the statement “…revealed 8 miRs down-regulated and 3 down regulated (Figure 3a)” is incomplete. What is the 3 down regulated referring to?
  3. In line 592, insert “of” after “because”.
  4. In Figures 3f-h, what is the dose of antagomir 139-5p used in the study?
  5. The authors should explain the relevance of miR scramble control used in their experiments in sections 3.2.1 and 3.2.2.
  6. In section 3.3.3, why was N1-S1 cells pretreated with 5-Aza before being inoculated into animals? My thought on this is that in clinical use of 5-Aza for HCC, the drug will be given after tumor has been diagnosed. To mimic what is done in the clinic, I would expect treating the orthotopic syngenic model with 5-Aza after tumor nodules have been observed.

Author Response

1) The authors did an extensive study to understand the mechanism of the antiproliferative agent, 5-aza. The authors in particular focused on the phenotypic and molecular effects of 5-aza, and they showed in both in-vitro and in-vivo studies that upregulation of miR 139-5P is important in the anticancer activity of 5-Aza in HCC. The study design and experiments are adequate, and the results are pretty convincing.

We wish to thank the reviewer for the appreciation of our experimental approach.

I recommend the manuscript for publication after the authors address the following points:

1)The authors should explain the significance of the Scratch test and cell migration experiments. Are these indicators of metastasis in HCC?

Yes, the scratch test as well as the FATIMA test and the Time-lapse analysis we used, are considered reliable tests to evaluate HCC cell migration, i.e. their metastatic potential. In this regard while HCC metastasis rarely involve extra liver tissues, they are mostly localized in the liver. To make this concept clearer, in the revised manuscript we have introduced in section 3.1.2 lines 533-535 the sentence: “this suggests that 5-Aza has the potential to down regulate the formation of the intra-hepatic metastasis occurring in HCC”.

2) In line 591, the statement “…revealed 8 miRs down-regulated and 3 down regulated (Figure 3a)” is incomplete. What is the 3 down regulated referring to?

In the revised manuscript, the mistake has been corrected (line 585) with the sentence: “…revealed 8 miRs down-regulated and 3 up regulated”.

3) In line 592, insert “of” after “because”.

In the revised manuscript, the mistake has been corrected.

4) In Figures 3f-h, what is the dose of antagomir 139-5p used in the study?

The dosage was the same used for the mir-139-5p, i.e. 100 nM; this information has been added to the legend of Figure 3.

5) The authors should explain the relevance of miR scramble control used in their experiments in sections 3.2.1 and 3.2.2.

miR scrambled was introduced as a negative control; it represents a control to evaluate the specific effects of miR 139-5p.

6) In section 3.3.3, why was N1-S1 cells pretreated with 5-Aza before being inoculated into animals? My thought on this is that in clinical use of 5-Aza for HCC, the drug will be given after tumor has been diagnosed. To mimic what is done in the clinic, I would expect treating the orthotopic syngenic model with 5-Aza after tumor nodules have been observed.

We perfectly agree with the reviewer that in the clinic 5-Aza is given only after tumor diagnosis. However, the rationale of our experiment in the paper was different. As we have shown in vitro that 5-Aza can reduce HCC cells adhesion, we wanted to confirm this observation in vivo, i.e. that HCC cell cannot adhere and graft in the liver when pre-treated by 5-Aza. Our data strongly indicate that indeed 5-Aza impairs the ability of HCC cells adhere in vivo in the liver and thus cannot give origin to tumor foci.

Reviewer 2 Report

In this study, the authors aim at understanding the mechanism of action of a demethylating drug, 5-Azacytidine. The authors propose that treatment with 5-Aza impairs HCC development by upregulating miR139-5p that results in the downregulation of ROCK2/cyclinD1/E2F1/CyclinB1 (pro-proliferative) pathway and Rock2/MMP-9 (pro-migratory) pathway. Prior to publication, the authors should clarify a number of important points as detailed below.

  • The rationale behind looking at the signaling pathway controlled by miR 139-5p is not clearly explained in the introduction and needs to be better connected in the introduction. For instance, has the effect of 5-Aza on miR 139-5p been previously reported in other tumors?

  • The effect of miR139-5p deregulation has been previously reported in HCC and other cancers and as such adds very minimal novelty. Since miR139-5p has several direct interacting targets, assessing the effect of 5-Aza induced up-regulation of miR139-5p on these reported targets through gene expression analysis experiments with 5-AZA treatment and miR139-5p inhibition experiments would be necessary. This will help understand if ROCK2 is integral in 5-Aza’s mechanism of action or if all the targets of miR139-5p have a similar effect. For instance,  the inverse regulation of  miR139-5p target KPNA2 has a similar effect on cell viability, cell cycle and migration in HCC.

  • In figure 1. 5-AZA at 30uM was tested for IHH and HUH-7 but not on the control cells. Including this would serve as a good comparison for the effect seen in HuH-7 at this concentration. Could the authors also include the data for the cell cycle analysis for these controls since the effect of 5-AZA on IHH and HuH-7 in cell viability seem similar?

  • The effect of 5-AZA on HCC cell viability has been previously reported and Figures 1a and b can perhaps be added as supplemental figures to support the cell cycle analysis data.

  • In section 3.2 authors need to correct the error in the statement: “In JHH-6, microarray analysis (protocol 1) revealed 8 miRs down-regulated and 3 down regulated (Figure 3a). “ Both are mentioned as down regulated

  • Could the authors include the expression of miR 139-5p for IHH cells upon treatment with 5-Aza as a comparison?

  • Figure 3b-e has been previously reported in HCC and adds very little novelty. This can be included in the supplemental data.

  • The mRNA levels of miR 139-5p after transfection in HuH7 seem to be lower than in JHH-6 cells yet, the ROCK2 protein level seems to be much lower in HuH-7. Could the authors comment on what could be causing such a varied effect between these cell lines? Also, the authors state that “miR 139-5p did not decrease ROCK2 mRNA (Figure 4g), suggesting a post-transcriptional regulation of ROCK2 levels as expected from a classical miR-mediated effect”  the levels of ROCK2 in JHH-6 clearly seem to be higher in miR 139-5p transfected cells. Statistics indicating significance need to be added to the figure and perhaps an explanation for these results.

  • For Figure 5, the dose of 5-Aza needs to be included. It is surprising that though there was a high overexpression of ROCK2 in HuH-7 cells (mRNA and protein overexpression) than in JHH-6 cells, ROCK2 expression does not seem to attenuate the anti-proliferative effect  of 5-Aza in these cells. Were JHH-6 and HuH-7 cells treated at the same 5-Aza concertation for this?

  • Figure 5b 5c 5d has been previously reported and doesn’t add any novelty. This can be moved to the supplemental file.

  • In figure 6e, the Cyclin B1 western blot figure is not convincing. Can a better western blot be included instead from the different experimental repeats?

  • What is the IC50 of 5-AZA in HuH7 cells? Was this considered for the in vivo 5-AZA concertation used?

  • The authors have shown the efficacy of 5-AZA across various in vivo models which is very much appreciated. Including data on a short term in vivo experiment by treating with 5-AZA using HCC cells with inducible ROCK2 expression can perhaps strengthen their data and provide validity for their findings. This is an important experiment and will add value to the in vitro findings.  The authors have demonstrated in vitro that treating HCC cells with 5-AZA results in downregulating ROCK2 protein levels which in turn results in reduced cell proliferation. Since, ROCK2 plays a pivotal role in 5-AZA mechanism of action, rescue experiments demonstrating this in vivo would add value.

Author Response

In this study, the authors aim at understanding the mechanism of action of a demethylating drug, 5-Azacytidine. The authors propose that treatment with 5-Aza impairs HCC development by upregulating miR139-5p that results in the downregulation of ROCK2/cyclinD1/E2F1/CyclinB1 (pro-proliferative) pathway and Rock2/MMP-9 (pro-migratory) pathway. Prior to publication, the authors should clarify a number of important points as detailed below.

1) The rationale behind looking at the signaling pathway controlled by miR 139-5p is not clearly explained in the introduction and needs to be better connected in the introduction. For instance, has the effect of 5-Aza on miR 139-5p been previously reported in other tumors?

 To fulfill the reviewer request, in the revised version of the manuscript we have included some sentences and novel references ([6-9]) addressing the effects 5-Aza on miR 139-5p in other tumors (lines 136-139). Additionally, we have made clear that miR 139-5p down regulation occurs, in addition to HCC, also in different other tumors (lines 97-99). Finally, we specified that epigenetic alterations contribute to carcinogenesis in different tumors beside HCC (line 87).

2) The effect of miR139-5p deregulation has been previously reported in HCC and other cancers and as such adds very minimal novelty.

The goal of our work was not to show that miR139-5p down-regulation occurs in HCC, rather that 5-Aza can restore miR139-5p expression in HCC. Moreover, we also aimed to elucidate the unknown molecular mechanisms responsible for the anti HCC effects of 5-Aza. This has been specified on lines 916-918.

3) Since miR139-5p has several direct interacting targets, assessing the effect of 5-Aza induced up-regulation of miR139-5p on these reported targets through gene expression analysis experiments with 5-AZA treatment and miR139-5p inhibition experiments would be necessary. This will help understand if ROCK2 is integral in 5-Aza’s mechanism of action or if all the targets of miR139-5p have a similar effect. For instance, the inverse regulation of  miR139-5p target KPNA2 has a similar effect on cell viability, cell cycle and migration in HCC.

Our data show that miR 139-5p transfection can recapitulate the 5-Aza effects (Supplement 4 g-l, Supplement 4 f, Figure 4 e-g). Moreover, in Figure 3 b-d we show that the anti mir 139-5p contrasts the effects of 5-Aza on cell viability and migration. Then, in Figure 4 e-f we show that miR 139-5p transfection down regulates the level of ROCK2. Finally, in Figure 5a we show that ROCK2 over expression contrasts 5-Aza effects on cell growth (JHH6). Together these observations support the connection between miR 139-5p and ROCK2 in the cellular models considered and support the functional role of miR 139-5p/ROCK2 in 5-Aza phenotypic effects. This has been specified on line 924-929. Said this, we cannot exclude that other 139-5p targets can contribute to the effects of 5-Aza. However, we concentrated on ROCK2, as this is a demonstrated player in HCC patients; other genes that could emerge from an in vitro analysis may not be necessarily relevant for the real world of HCC patients.

4a) In figure 1. 5-AZA at 30uM was tested for IHH and HUH-7 but not on the control cells. Including this would serve as a good comparison for the effect seen in HuH-7 at this concentration.

In the original manuscript, the dose of 30 mM 5-Aza was not tested in IHH, which are non-tumorigenic in vivo although being immortalized. In the revised manuscript, this novel data has been included in supplement 2 a-b. At this concentration, the effect on IHH is comparable to that of HuH7. Because of this, in the revised manuscript we have re-modulated the original sentence: “In IHH cells [47, 52], and even more in hESC-derived hepatocyte-like cells (HLC) [53], introduced as controls for normal hepatocytes, the 5-Aza effects were less pronounced (Figure 1a,b)” as it follows:” 5-Aza effects, also detectable in the non-tumorigenic but immortalized IHH cells [47, 52], were instead minimal in HLC [53], introduced as controls for normal hepatocytes (Figure 1c, Supplement 2 a,b)”. lines 523-525. It should be noted that the mechanisms responsible for the decrease of IHH viability might be different from those of JHH-6/HuH-7. Indeed, in IHH 5-Aza induces an accumulation in G2/M but not in G1/G0 (Figure 1c) while in JHH-6 and HuH-7 accumulation both in G2/M and in G1/G0   are observed (Figure 1a-b).

4b)Could the authors also include the data for the cell cycle analysis for these controls since the effect of 5-AZA on IHH and HuH-7 in cell viability seem similar?

In the original manuscript, cell cycle data have not been reported for IHH. In the revised manuscript, this novel data has been included in Figure 1c. In this case, at 6 mM we just observe an accumulation of cells in G2/M but not in G1/G0 as instead detected in JHH-6/HuH-7. Thus the mechanisms of 5-Aza might be different in the non-tumorigenic IHH compared to the tumorigenic JHH-6 and HuH-7 cells. This observation is in line with the fact that 5-Aza does not increase miR 139-5p expression in IHH (novel supplement 4a).

5) The effect of 5-AZA on HCC cell viability has been previously reported and Figures 1a and b can perhaps be added as supplemental figures to support the cell cycle analysis data.

According to referee request, cell number and viability data have been moved to supplementary material 2 a-b.

6) In section 3.2 authors need to correct the error in the statement: “In JHH-6, microarray analysis (protocol 1) revealed 8 miRs down-regulated and 3 down regulated (Figure 3a). “ Both are mentioned as down regulated

In the revised manuscript, the mistake has been corrected.

7) Could the authors include the expression of miR 139-5p for IHH cells upon treatment with 5-Aza as a comparison?

In the revised manuscript, this data has been added in supplement 4a. Interestingly, miR 139-5p is not upregulated by 5-Aza in IHH. This suggests that miR 139-5p upregulation is a phenomenon restricted to tumorigenic HCC cells. In the revised manuscript, this observation has been reported on lines 592-593.

8) Figure 3b-e has been previously reported in HCC and adds very little novelty. This can be included in the supplemental data.

 According to referee request, the data of Figure 3 b-e have been moved to supplementary material 4 g-l

9a) The mRNA levels of miR 139-5p after transfection in HuH7 seem to be lower than in JHH-6 cells yet, the ROCK2 protein level seems to be much lower in HuH-7. Could the authors comment on what could be causing such a varied effect between these cell lines?

Whereas many unpredictable factors can concur to explain this difference, it is possible that in JHH-6 the mir139-5p amount after transfection have saturated RISC complex. In this regards, we do not know how abundant RISC is in JHH-6 and in HuH-7. Additionally, the difference may depend on a dissimilar half-life (stability) of ROCK2 in JHH-6 and in HuH-7, as the consequence of diverse post-translational modifications occurring in the two cell lines. Despite these considerations, our main aim was to show that in both cell lines miR139-5p transfection could reduce ROCK2 protein level. Notably, the different decrease of ROCK2 in the two cell lines is of about 20%, so not particularly dramatic and thus, it should not affect the phenotypic results we observed.

9b) Also, the authors state that “miR 139-5p did not decrease ROCK2 mRNA (Figure 4g), suggesting a post-transcriptional regulation of ROCK2 levels as expected from a classical miR-mediated effect” the levels of ROCK2 in JHH-6 clearly seem to be higher in miR 139-5p transfected cells. Statistics indicating significance need to be added to the figure and perhaps an explanation for these results.

According to reviewer request, in the revised manuscript the statistic has been introduced in figure 4g. The reasons for the increase of ROCK2 mRNA levels are not fully clear. It is possible to speculate that the reduction in the protein level of ROCK2 in JHH-6 triggers a “reactive” tentative to contrast the decrease of the protein by pushing the transcriptional process. In the revised manuscript, this consideration has been introduced on lines 629-631. Despite this, the concept of the post-transcriptional regulation of ROCK2 levels by miR 139-5p, remains valid.

10a) For Figure 5, the dose of 5-Aza needs to be included.

In the revised manuscript, the 5-aza concentrations have been added in the legend of Figure 5.

10b) It is surprising that though there was a high overexpression of ROCK2 in HuH-7 cells (mRNA and protein overexpression) than in JHH-6 cells….,

The data reported in supplement 5c indicate that ROCK2 protein levels are higher in JHH-6 compared to HuH7 following pROCK2 transfection. The opposite occurs at the mRNA level; however, we believe that for the phenotypic effects the protein levels are more relevant.

10c)..…ROCK2 expression does not seem to attenuate the anti-proliferative effect  of 5-Aza in these cells. Were JHH-6 and HuH-7 cells treated at the same 5-Aza concertation for this?

JHH-6 and HuH-7 have been treated with 0,8 and 6 mM 5-aza, respectively;  the dosage was higher in HuH-7 due to their increased resistance to 5-Aza effects. We did not use the highest dose for JHH-6 (6 mM) and HuH-7 (30 mM) employed in other experiments, as the combination with transfection resulted to be too toxic for the cells. Transfection is per se responsible for a significant unspecific toxicity and as we have optimized the transfection procedure for the two cell lines in previous works, we could reduce the overall toxicity (unspecific from transfection and specific from 5-Aza) only by reducing 5-Aza dosage. This approach resulted to be effective in JHH-6 where we could clearly detect the anti 5-Aza effect of ROCK2 overexpression. In the case of HuH-7, we believe that the unspecific toxicity due to transfection, in part masked the pro-proliferative effect of ROCK2 overexpression and eventually the anti 5-Aza effect in comparison to EGFP transfected cells. In addition, ROCK2 protein levels were lower in HuH-7, a fact that may have concurred to make its overexpression less effective in contrasting the anti 5-Aza effects. Despite this consideration, our data indicate the tendency of ROCK2 to contrast 5-Aza effect on HuH-7 growth.

11) Figure 5b 5c 5d has been previously reported and doesn’t add any novelty. This can be moved to the supplemental file.

 In the revised manuscript, these data have been moved to supplement material 5 h-l.

12) In figure 6e, the Cyclin B1 western blot figure is not convincing. Can a better western blot be included instead from the different experimental repeats?

In the revised manuscript, a novel blot for cyclin B1 has been introduced in Figure 7e.

11) What is the IC50 of 5-AZA in HuH7 cells?

The IC50 calculated on the number of cells remaining after 5-Aza treatment is of 22,3 mM.

12) Was this considered for the in vivo 5-AZA concertation used?

For the in vivo experiments, we considered the dose of 40 mM of 5-Aza. As reported in the text in section 3.3.1, lines 810-813, the reason was that: “This amount of 5-Aza (corresponding to an estimated initial intra-tumor concentration of 40 mM) was chosen to be very similar to the most effective concentration used in vitro for HuH-7 (30 mM)”. This in vivo dosage was slightly higher than that in vitro as we considered the possibility that a certain drug drainage via the blood vessel could have reduced the effective amount in the tumor. Additionally, we have also tested a dosage of 20 mM (comparable to the IC50) of 5-Aza showing that it was also effective in reducing tumor growth in vivo. Thus, we believe we have fully considered the in vitro data to set up the in vivo tests.

13a) The authors have shown the efficacy of 5-AZA across various in vivo models which is very much appreciated.

We thank the reviewer for the appreciation of our in vivo experimental approach.

13b) Including data on a short term in vivo experiment by treating with 5-AZA using HCC cells with inducible ROCK2 expression can perhaps strengthen their data and provide validity for their findings. This is an important experiment and will add value to the in vitro findings.  The authors have demonstrated in vitro that treating HCC cells with 5-AZA results in downregulating ROCK2 protein levels which in turn results in reduced cell proliferation. Since, ROCK2 plays a pivotal role in 5-AZA mechanism of action, rescue experiments demonstrating this in vivo would add value.

We thank the reviewer for having raised this important aspect. Although we thought about this experiment, for technical reasons we were not able to perform it.  From the transient transfection performed in vitro to overexpress ROCK2, we observed that the expression of ROCK2 could last at maximum 7 days from transfection. Considering that the tumor appears in mice 3-4 weeks from HuH-7 injection, it is clear that the transiently transfected HuH-7 could not have retained the ability to overexpress ROCK2 after such a long time from injection in vivo. We therefore tried to establish a stable overexpression of ROCK2 by mean of a retroviral construct in HuH-7. Unfortunately, however, the stable overexpression of ROCK2 resulted to be toxic for HuH-7 as all the clones we selected died after some days from the procedure. We have not tried the same procedure for JHH-6 has these cells do not graft in mice. Therefore, despite being a very interesting experiment, we cannot perform it.